# Anatomically inspired digital twins capture hierarchical object representations in visual cortex

**Emanuele Luconi**[†‡]     **Dario Liscai**[†‡]

**Carlo Baldassi**[‡]     **Alessandro Marin Vargas**[§]     **Alessandro Sanzeni**[‡*]

[‡]Department of Computing Sciences and BIDSA, Bocconi University, Milan, Italy
[§]Stanford University, USA    [†]Equal contribution

## Abstract

Invariant object recognition–the ability to identify objects despite changes in appearance–is a hallmark of visual processing in the brain, yet its understanding remains a central challenge in systems neuroscience. Artificial neural networks trained to predict neural responses to visual stimuli ("digital twins") could provide a powerful framework for studying such complex computations in silico. However, while current models accurately capture single-neuron responses within individual visual areas, their ability to reproduce how populations of neurons represent object identity, and how these representations transform across the cortical hierarchy, remains largely unexplored. Here we examine key functional signatures observed experimentally and find that current models account for hierarchical changes in basic single-neuron properties, such as receptive field size, but fail to capture more complex population-level phenomena, particularly invariant object representations. To address this gap, we introduce a biologically inspired hierarchical readout scheme that mirrors cortical anatomy, modeling each visual area as a projection from a distinct depth within a shared core network. This approach significantly improves the prediction of population-level representational transformations, outperforming standard models that use only the final layer, as well as alternatives with modified architecture, regularization, and loss function. Our results suggest that incorporating anatomical information provides a strong inductive bias in digital twin models, enabling them to better capture general principles of brain function.

## 1   Introduction

Invariant object recognition refers to the ability to recognize objects despite changes in position, shape, scale, or luminance. Despite its importance, understanding the neural basis of this computation, and of other complex visual functions, remains a central challenge in systems neuroscience (Kar & DiCarlo, 2024). Anatomical studies have revealed a clear hierarchical organization of visually responsive areas in the brain across species (DiCarlo et al., 2012; Harris et al., 2019). In mice, this hierarchy extends from the retina through the thalamus to the primary visual cortex (V1), the lateromedial area (LM), and other higher visual areas (including RL and AL), as inferred from anatomical connectivity (Fig. 1A). Functional studies have shown that this anatomical hierarchy is accompanied by systematic changes in single-neuron response properties (Siegle et al., 2021; Glickfeld & Olsen, 2017) and by increasingly linearly decodable object representations (Froudarakis et al., 2020; Hoeller et al., 2024).

Despite significant progress, fundamental questions remain about the neural mechanisms underlying invariant object recognition. What specific transformations in neural representations enable this

---

[*]Corresponding author: `alessandro.sanzeni@unibocconi.it`

39th Conference on Neural Information Processing Systems (NeurIPS 2025).

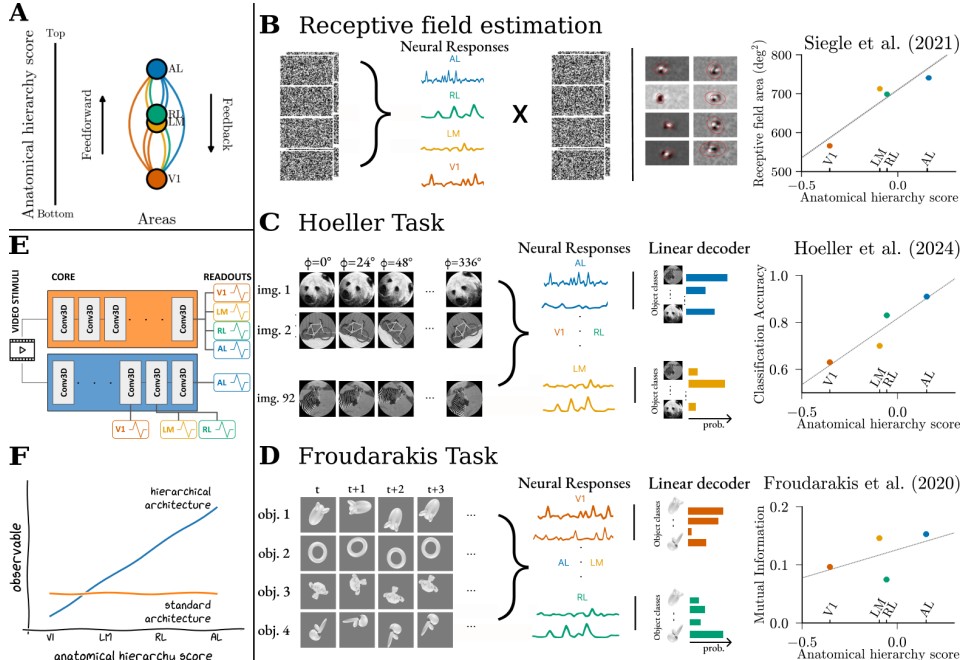

Figure 1: **Probing cortical hierarchy in the mouse visual system.** (**A**) Hierarchy scores for four visual areas (V1, LM, RL, AL) derived from anatomical connectivity (Harris et al., 2019). The method analyzes laminar termination patterns of inter-areal axonal projections to assign each area a continuous position in a global cortical hierarchy. While the original analysis included over ten areas, we focus on those present in the MICrONS dataset (Consortium et al., 2021). Scores follow the trend V1<LM ≲RL<AL. (**B**) Receptive field (RF) sizes estimated from neural responses to visual stimuli. RF size increases approximately monotonically with anatomical hierarchy (V1<RL≲LM<AL) (Siegle et al., 2021). (**C**) Rotation-invariant object recognition (Hoeller et al., 2024) assessed using static images at different rotations, recording neural responses, training a support vector machine (SVM) and testing it on unseen rotations. Classification accuracy increases monotonically with hierarchy (V1<LM<RL<AL). (**D**) Invariant object recognition (Froudarakis et al., 2020) assessed by presenting four moving objects undergoing identity-preserving transformations. Discriminability increases with hierarchy, except for RL, which performs poorly (RL < V1 < LM < AL). (**E**) Schematic of standard digital twin models (top), where a core network generates a multi-layer stimulus embedding and the final layer is used to predict neural responses. In our anatomically inspired architecture (bottom), each area is predicted from a different layer of the core, selected based on its anatomical hierarchy score. (**F**) While both architectures capture RF size increase across areas (as in panel B), only the hierarchical architecture reproduces the increase in invariant object recognition along the hierarchy (panels C–D).

process, and how do they vary across cortical areas? Experimentally answering these questions would require dense sampling of neural responses to high-dimensional sets of object transformations, a task currently infeasible due to the limited recording time available in behaving animals. A promising alternative is offered by *digital twins*: artificial neural networks trained to predict neural responses to visual stimuli (Klindt et al., 2017; Ecker et al., 2018; Sinz et al., 2018; Lurz et al., 2020; Wang et al., 2025). These models have recently emerged as powerful tools for characterizing the functional properties of the visual system and have been shown to replicate diverse single-neuron response features (Pogoncheff et al., 2023; Walker et al., 2019; Bashiri et al., 2025; Tong et al., 2023; Wang et al., 2025; Ustyuzhaninov et al., 2022; Ding et al., 2023). However, most existing models focus on predicting single-neuron responses without explicitly assessing whether they capture population-level properties, such as differences in representational geometry across visual areas which is critical for invariant object recognition. Without this capability, their ability to reveal how neural circuits progressively construct more abstract visual representations is significantly constrained.

Here, we address this gap by evaluating the ability of digital twins of the mouse visual cortex to reproduce experimental findings on how object representations evolve across the visual cortical hierarchy. We focus on three key experimental results, each probing different aspects of the trans-

formation from low-level to high-level representations. First, we test whether models capture the increase in receptive field size across visual areas, a well-established signature of increasing spatial integration that supports invariance to object position and scale (Siegle et al., 2021) (Fig. 1B). Second, we evaluate whether models capture the increase in rotation-invariant object decoding across the visual hierarchy, as demonstrated by Hoeller et al. (2024). In their study, linear support vector machines (SVMs) were trained on neural activity from different areas to classify object identity under unseen rotations (Fig. 1C). Third, we assess whether the models capture the progression of object discriminability across areas, as discovered by Froudarakis et al. (2020). In this task, neural responses were recorded as mice viewed four objects undergoing identity-preserving transformations, including rotation, scaling, translation, and variations in illumination conditions (Fig. 1D).

We find that current digital twins are able to replicate low-level functional properties, such as receptive field size gradients across areas, but fail to account for differences in higher-level representational invariances (Fig. 1E,F). In particular, they do not reproduce the experimental findings of Hoeller et al. (2024) or Froudarakis et al. (2020) when tested under analogous conditions. To address this limitation, we introduce a biologically inspired modification to the model architecture. Conventional approaches predict responses across all brain areas using a shared set of visual features, referred to as the "core network" (Fig. 1E). However, this design overlooks the distinct computational transformations applied within different visual regions. Instead, we propose a hierarchical readout (Fig. 1E), where neural responses corresponding to different brain areas are mapped to distinct layers within the core network rather than extracted from a single shared final layer. This architectural change introduces an inductive bias that improves the alignment between in-silico and experimentally-observed population-level representations across the hierarchy (Fig. 1F) while also improving single-neuron predictions in higher visual areas. We show that mimicking the anatomical organization is key to achieve this alignment, whereas other alternatives modifications to the baseline architecture fail to capture the experimental phenomenology. Our work provides the first demonstration that digital twins can be adapted to reproduce representational transformations across mouse visual areas, and offers a framework for using such models to investigate the mechanisms underlying behaviorally relevant visual computations.

We summarize our main contributions as:

- We simulated different experiments to probe hierarchical representations in digital twins
- We show that current models fail to replicate functional hierarchies across visual areas in multiple experimental datasets (Froudarakis et al., 2020; Hoeller et al., 2024)
- We investigate different architectures and training framework
- We propose a new biologically-inspired hierarchical readout mechanism that captures experimental results.

## 2 Related work

**Hierarchical processing in the visual cortex.** The visual system exhibits a hierarchical organization where information flows through a network of interconnected areas (Seabrook et al., 2017; Siegle et al., 2021). In mice, experimental evidence (de Vries et al., 2018; Consortium et al., 2021; Siegle et al., 2021) has revealed substantial differences between primary visual cortex (V1) and higher-order areas (LM, RL, AL) in receptive field properties (Siegle et al., 2021; Glickfeld & Olsen, 2017), spatiotemporal characteristics (Murakami et al., 2017; Piasini et al., 2021), and critically, invariant object recognition capabilities (Froudarakis et al., 2020; Hoeller et al., 2024). These hierarchical transformations provide a critical baseline for evaluating computational models of visual processing and motivate our investigation into whether digital twins can not only predict neural responses but also capture the fundamental representational changes across cortical areas.

**Modeling hierarchical representations with digital twins.** Deep neural networks have emerged as powerful "digital twins" for modeling neural responses to visual stimuli, following two main approaches: task-driven models optimized for visual tasks (Yamins et al., 2014; Bakhtiari et al., 2021; Kubilius et al., 2019; Zhuang et al., 2021; Pogoncheff et al., 2023), and data-driven models trained directly on neural recordings (Klindt et al., 2017; Cadena et al., 2019; Lurz et al., 2020; Safarani et al., 2021; Willeke et al., 2022; Li et al., 2023; Wang et al., 2025). Despite their success in predicting single-neuron responses (Turishcheva et al., 2024a; Xu et al., 2024), these models struggle to capture the hierarchical organization of representations across the visual system (St-Yves et al., 2023; Dyballa

et al., 2024; Liscai et al., 2025). In primates, task-driven networks trained on object recognition revealed a clear hierarchy along the ventral stream, with early layers predicting V1 and deeper layers aligning with V4 and IT (Yamins et al., 2014). In mice, however, no such progression was found: ImageNet-trained and even randomly initialized networks performed similarly across V1, LM, AL, and RL (Cadena et al., 2019), and although unsupervised objectives improved predictions and mapped V1 to early layers and higher areas to intermediate layers, they still failed to separate higher-order regions (Nayebi et al., 2023). This limitation highlights the need for architectural innovations that better align with the biological organization of the visual cortex.

**Single-neuron and population-level analysis.** Traditionally, neural network models in visual neuroscience have focused on single-neuron responses, investigating how individual neurons encode sensory stimuli. Studies have examined feature selectivity in V1 by finding the most exciting inputs (Walker et al., 2019), phase-invariant and spatial encoding in V1 (Ding et al., 2023; Ustyuzhaninov et al., 2022; Bashiri et al., 2025), and the columnar organization of selectivity in V4 (Willeke et al., 2023; Burg et al., 2024). Despite these advances, single-neuron analyses fail to capture the broader hierarchical structure of population-level representations across visual areas. Pogoncheff et al. (2023) investigated how population structure influences V1 predictability, while Margalit et al. (2024) examined topological differences between early and higher-order visual regions. Expanding on this, Liscai et al. (2025) analyzed the geometry of neural representations, showing that regularization improves the differentiability of representations but still fails to replicate experimentally observed patterns of discriminability across visual areas. To address these limitations, we introduce a biologically-inspired hierarchical readout mechanism that maps different network depths to distinct visual areas, significantly improving the model's ability to capture both single-neuron and population-level responses.

## 3 Digital twins overlook differences in object representations across areas

**Digital twin of the mouse visual cortex.** We modeled neuronal responses using a compact digital twin with state-of-the-art performance that includes a core module, a readout module, a behavioral module, and a shifter module (see Appendix A for details). The core component consists of a four-layer 3D Convolutional Neural Network (CNN), which transforms input videos into a high dimensional embedding. The behavioral module processes behavior data and transforms it into a latent representation which is stacked to the core's output. The readout module, adapted from Lurz et al. (2020), translates this encoded representation into predictions of individual neuronal activity belonging to different brain areas. Its functionality is organized into two key components: a spatial mask, which defines receptive field locations, and a set of feature weights that linearly aggregate information from the final core layer to produce frame-by-frame neural response predictions. Additionally, the model incorporates a shifter module, which processes pupil center coordinates to estimate gaze-related shifts in receptive field positioning. This transformation is applied linearly across all neurons and is finalized with an ELU+1 activation function to ensure non-negative outputs. We refer to the configuration with $N = 4$ layers in the core and standard readout as the *baseline model* throughout the paper. We trained the baseline model, as well as of all variants discussed in the following sections, by minimizing the Poisson loss between predicted and recorded responses. Training was performed on 83,222 units from V1, 14,817 from LM, 12,599 from RL, and 4,734 from AL, using data from the MICrONS dataset (Consortium et al., 2021), recorded via two-photon calcium imaging in response to grayscale natural movies. Further details on the model architecture, training procedures, and evaluation metrics are provided in Appendix A.

Our baseline digital twin model achieved strong single-neuron prediction accuracy, comparable to previous work (Wang et al., 2025). Prediction performance was highest in V1, with moderate reductions observed in LM, RL, and AL (Fig. 2A). This trend likely reflects both the inherent variability in higher-order areas (de Vries et al., 2018) and the larger number of V1 neurons, which might have biased the model toward optimizing this area compared to the others.

**Baseline model fails to capture how object representations evolve across the visual cortical hierarchy.** To assess whether digital twins of the mouse visual cortex capture functional distinctions across areas, we evaluated whether our baseline model reproduces key experimental findings summarized in Fig. 1B-D. Specifically, we replicated in silico the experimental protocols used to generate these measurements and compared model-predicted responses to their biological counterparts. Details of the in silico procedures are provided in Appendix B.

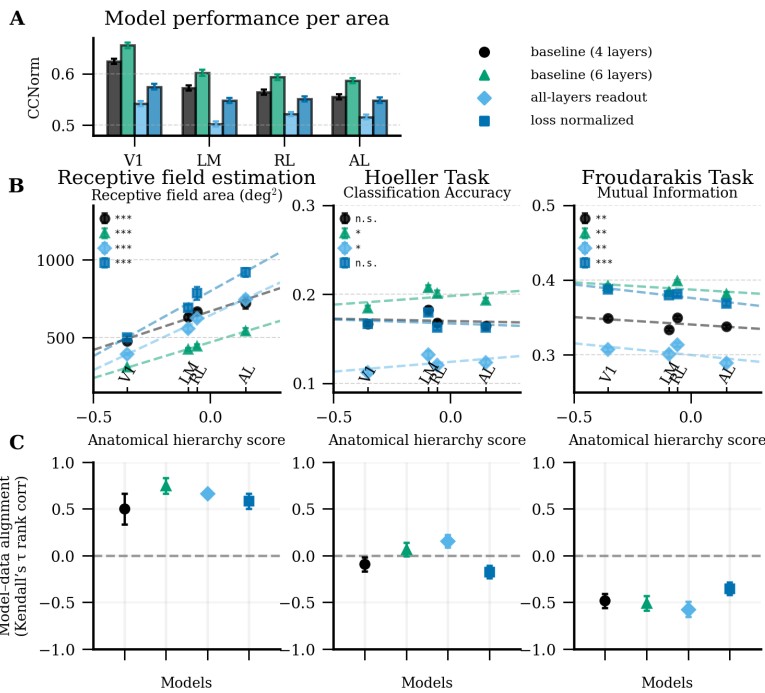

Figure 2: **Digital twin fails to capture hierarchical transformations in object representations across visual areas.** (**A**) Single-neuron prediction performance across areas for the baseline model and its variants. Bars indicate mean±SEM across training seeds, as in subsequent analogous plots. (**B**, left to right) RF size as a function of anatomical hierarchy score (dots) and best linear fit (dashed line) for each model; decoding accuracy for SVMs trained on neural representations of the digital twin model presented with rotated images from the Hoeller task; and object discriminability based on neural representations of the digital twin model presented with videos from the Froudarakis task. Here and in subsequent analogous plots: markers and errorbars represent mean and SEM across recording sessions and training seeds; the legend indicates if the slope of the linear fit is significantly different from zero (Wald Test): $p < 0.001$ (***), $p < 0.01$ (**), $p < 0.05$ (*), $p > 0.05$ (n.s.). (**C**) Agreement between model-predicted and experimentally observed orderings in (B), as quantified by Kendall's $\tau$ rank correlation coefficient, which measures the correspondence between two rankings based on the number of concordant and discordant pairs (see definition in Appendix A.3.1).

First, we evaluated whether the units of the baseline model exhibit systematic increases in receptive field (RF) size along the cortical hierarchy, as observed in Siegle et al. (2021). To this end, we computed RFs by presenting spatiotemporal white noise and applying standard spike-triggered averaging (See appendix B.1). To quantify RF size, we fit a two-dimensional Gaussian to the RF map of each unit and defined size as the product of the standard deviations along the two principal axes. RF size systematically increased along the model's cortical hierarchy (Fig. 2B), aligning well with experimental observations (Fig. 1B). This result is notable: despite all area-specific units being projections from a shared core representation, the model trained on natural stimuli implicitly captured topographic distinctions across areas. We further quantified the agreement using Kendall's $\tau$, a rank correlation metric that compares pairwise area relationships (e.g., whether the RF size in LM exceeds that in V1, and so on). Kendall's $\tau$ provides a measure of the consistency between the experimentally observed functional hierarchy for the receptive field size (V1<RL≲LM<AL, Fig. 1B) and the one predicted by the model, where a value of $+1$ indicates perfect agreement, whereas a Kendall's $\tau$ of $-1$ complete disagreement. The baseline model achieved a value of 0.50, indicating moderate consistency (Fig. 2C). The main difference was due to the predicted receptive field size in LM, which, unlike in the data, was not significantly larger than the one in RL.

Next, we examined whether the digital twin supports rotation-invariant object recognition and how this capacity evolves across cortical areas. Following Hoeller et al. (2024), we implemented a

classification task with 92 static objects at different rotations. A linear SVM was trained on the model-predicted neural responses to identify object identity across 92 classes for held-out rotations (details in Appendix B.2). As shown in Fig. 2B, classifiers trained on neural responses performed significantly above chance, demonstrating that object identity information was preserved across rotations. However, classification accuracy did not exhibit a systematic trend with anatomical hierarchy. Kendall's $\tau$ was negative (Fig. 2C), indicating that the area-to-area relationships predicted by the model deviated from biological data (Fig. 1C)–for instance, performance in AL was lower than all other areas. To further challenge the model, we introduced a more complex task inspired by Froudarakis et al. (2020), incorporating variations in object rotation, scale, position, and lighting (details in Appendix B.3). While object identity remained decodable above chance, classification accuracy again failed to correlate with hierarchical level (Fig. 2B). The negative Kendall's $\tau$ (Fig. 2C) confirmed that model area-to-area performance orderings did not match biological observations (Fig. 1D).

Taken together, these findings reveal a critical shortcoming of the baseline digital twin. While the model successfully captured low level aspects of the cortical hierarchy, such as single neuron responses and the systematic increase in receptive field size across areas, it failed to reproduce the functional transformations necessary for invariant object recognition. In particular, the model did not exhibit the progressive stabilization of object identity across areas that is characteristic of the biological visual system.

## 4  Architectural and training modifications do not induce hierarchical representations

**Rebalancing visual areas during training.** Like the mouse brain, the MICrONS dataset includes many more neurons from V1 than from higher visual areas. This imbalance could bias the digital twin toward V1-dominated representations due to the shared core, potentially explaining its failure to capture transformations across areas. Supporting this, the model showed higher single-neuron prediction accuracy in V1 (Fig. 2A). To test whether this imbalance affected hierarchical organization, we trained a model variant with a reweighted loss that scaled each area's contribution by the inverse of its neuron count (details in Appendix A.4), ensuring equal influence during training. The reweighted model trained successfully and produced more uniform accuracy across areas, but this was due to reduced performance in V1, with no improvement observed in other areas (Fig. 2A). We then tested whether this modification improved the model's ability to capture functional hierarchy. Receptive field sizes remained consistent with experimental observations, indicating that spatial tuning properties were preserved (Fig. 2B,C). However, the hierarchical structure of object representations remained unchanged. On the Hoeller task (Hoeller et al., 2024), the model still failed to reproduce the correct order across areas (Fig. 2B,C). Similarly, on the Froudarakis task (Froudarakis et al., 2020), the hierarchy was again reversed, with AL and LM performing below V1 and only minor differences across areas (Fig. 2B,C). These results suggest that reweighting the training loss is not sufficient to induce the transformations in object representations observed experimentally across the visual hierarchy.

**Readout with access to all layers features.** In the baseline model, all neurons read out from the final core layer, regardless of cortical area. However, both cortical anatomy and physiology suggest that lower (higher) areas rely on simpler (more complex) features, paralleling how feature complexity increases with depth in convolutional networks (Goodfellow et al., 2009; Zeiler & Fergus, 2014; Cohen et al., 2020). Therefore, V1 neurons may be better modeled using earlier layers, while higher areas could benefit from deeper representations. To test this hypothesis, we developed a variant of the baseline model in which the readout for each neuron was computed as a projection of kernel activations across multiple layers, rather than from the final layer alone (details in Appendix A.4). Despite its biological plausibility, and the fact that it represents a generalization of the baseline, this model underperformed relative to the baseline, with a substantial decline in single-neuron prediction accuracy (Fig. 2A). This is likely due to the increased model complexity which made optimization more difficult and led to overfitting, despite an L1 regularization term being added to the loss. Receptive field size increased across cortical areas relative to the baseline model (Fig. 2B,C), with changes that were qualitatively similar but larger in magnitude. On both Hoeller et al. (2024) and Froudarakis et al. (2020) tasks, object discriminability decreased across all areas, and differences between areas remained minimal (Fig. 2B,C). These results suggest that, at least given current data

constraints, expanding the readout to include features from multiple layers in an unstructured fashion does not enhance the model's ability to capture area-specific functional organization in the visual cortex.

**Increasing model depth.** Another limitation of the baseline model is its shallow core, only four convolutional layers, which may constrain feature complexity and hinder its ability to capture transformations across cortical areas. To evaluate this, we trained a series of models with increasing depth, keeping all other architectural components fixed (details in Appendix A.4). Increasing the number of layers from four to six yielded improvements in single-neuron predictive accuracy across all areas (Fig. 2A). However, adding layers beyond six did not yield consistent additional improvements (Fig. 3A), suggesting a saturation point in representational benefits under current data constraints. Receptive field properties in this model variant showed a similar degree of agreement with experimental data as observed in the other models discussed above (Fig. 2B,C). For the functional hierarchy, we found that the six-layer model showed improved object-level discriminability relative to the baseline across all areas, along with a mild improvement in capturing inter-area differences (Fig. 2B,C). Although the six-layer model assigned higher discriminability to LM, RL, and AL than to V1, partially recovering the experimentally observed ordering for the Hoeller task, it still misranked AL below LM. In the Froudarakis task, performance remained flat: V1 continued to exhibit greater discriminability than higher-order areas, contrary to physiological findings. Together, these results suggest that increased depth improves the overall quality of object representations, likely by enabling the learning of higher-order visual features. Yet, as shown in Fig. 2B,C, this improvement is not sufficient to reproduce the known functional hierarchy observed in mouse visual cortex.

**Evaluation of a state-of-the-art large-scale model.** The foundation model of Wang et al. (2025) represents the current state of the art in mouse visual cortex modeling, with substantially more parameters (Table 1), neurons, and training data than typical architectures. We evaluated this model in our population-level framework to test whether scaling alone is sufficient to reproduce hierarchical organization (Supp. Fig. S3). At the single-neuron level, the model from Wang et al. achieved strong predictive accuracy (median CCnorm across sessions: [0.58-0.76]). At the population level, it captured the broad distinction between V1 and higher areas but failed to reproduce finer differences across higher-order regions. In both the Hoeller (Hoeller et al., 2024) and Froudarakis (Froudarakis et al., 2020) tasks, AL, RL, and LM were predicted to perform nearly identically, inconsistent with experimental data showing a clear AL advantage (Supp. Fig. S3). These results suggest that larger and more complex architectures, even when trained on more data, do not by themselves recover the functional hierarchy of mouse visual cortex.

## 5 A biologically inspired hierarchical readout capture differences across visual areas

**Hierarchical readout improves performance and functional hierarchy.** The baseline model assumes that neural responses across the visual cortex can be described as projections from a shared high-dimensional embedding of visual stimuli, extracted from the final layer of the core. However, this assumption is inconsistent with known anatomical and functional features of the visual system. For instance, V1 is the primary driver of LM (Fig. 1A), suggesting that LM's neural representation is not just an independent projection of a shared embedding but a transformed version of V1 representations. A similar principle applies to other visual areas, like RL and AL, which are positioned differently within the visual hierarchy (Fig. 1A). These observations motivate an alternative digital twin architecture, in which the core network progressively transforms visual stimuli through a sequence of layers, and neurons from each area are read out from different depths along this transformation, consistent with their anatomical positions. We refer to this architecture as the *hierarchical readout model* (details in Appendix A.4). As previously noted, single-neuron prediction accuracy in the baseline model increased with depth up to six layers and then plateaued (Fig. 2A and Fig. 3A). We therefore used an 8-layer core to ensure V1 was read out at its optimal depth (layer 6), with readouts for LM and RL assigned to the same layer (layer 7) due to their similar anatomical positions, and AL to layer 8, consistent with its hierarchical position (Fig. 1A).

The hierarchical readout model trained successfully and achieved single-neuron prediction accuracy that matched or exceeded that of the previously tested architectures (Fig. 2A and Fig. 3A). Notably, unlike the baseline 8-layer model with readouts from the final layer alone, the hierarchical model showed improved performance in predicting responses in AL compared to the 6-layer baseline. This

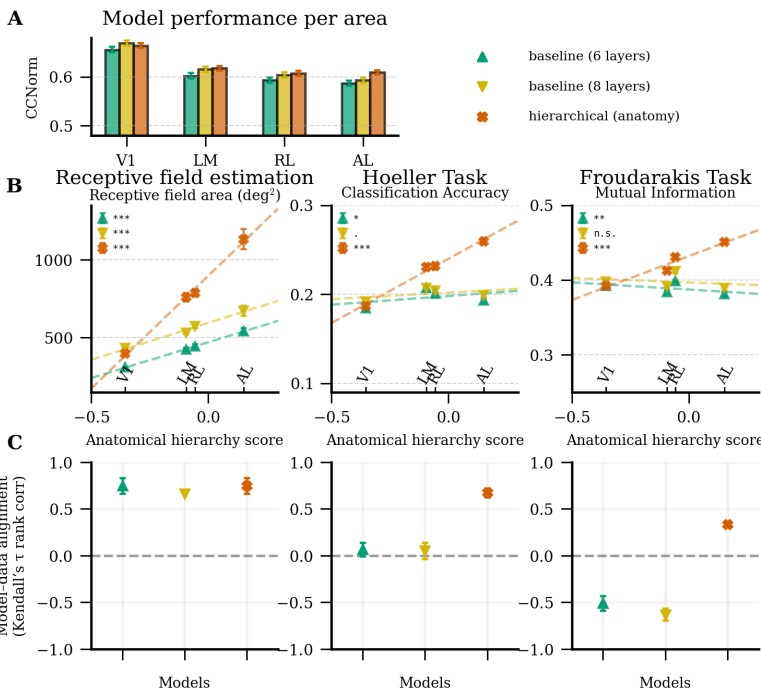

Figure 3: **The hierarchical model captures gradient in invariant object classification along visual areas.** (**A**) Single-neuron prediction performance across areas for the baseline model (6 or 8 layers) and the hierarchical model. In baseline models (top), neural responses are predicted from the final layer of a shared core network. In the hierarchical model (bottom), each area is predicted from a different layer, selected based on its anatomical hierarchy score. (**B**, left to right) RF size, classification accuracy in the Hoeller task, and object discriminability in the Froudarakis task, each plotted as a function of anatomical hierarchy score (dots) with best linear fits (dashed lines), shown separately for the baseline and hierarchical models. (**C**) Agreement between model-predicted and experimentally observed orderings in (B), as quantified by Kendall's $\tau$ rank correlation coefficient.

suggests that distributing readouts across layers in accordance with anatomical hierarchy facilitates the learning of area-specific neural representations. This architectural change also improved the model's ability to capture the functional hierarchies observed experimentally in receptive field size (Fig. 3B,C) and object recognition tasks (Fig. 3B,C) . In the hierarchical readout model, receptive field size increased monotonically along the cortical hierarchy, in agreement with experimental data. In both the Hoeller and Froudarakis tasks, the hierarchical model not only achieved the highest recognition accuracy across all tested models (Fig. 3B,C) but also exhibited a gradient of decoding performance across areas that mirrored experimental findings (Fig. 3B,C): accuracy increased from V1 to RL and LM, peaking in AL. Interestingly, the model placed RL in an intermediate position (between AL and LM) in both discrimination tasks. This result is consistent with Hoeller et al.'s findings, but at odds with Froudarakis et al.'s results. We were unable to reconcile this discrepancy and will address it in the discussion.

**Aligning readout structure with anatomical hierarchy improves functional predictions.** We next asked whether the specific anatomical ordering of hierarchical readouts was critical for the observed improvements. In principle, the better alignment could simply arise from distributing the readouts across different layers, rather than placing them all in the final layer, as in the baseline 8-layer model, thereby increasing representational flexibility, without requiring alignment to the actual cortical hierarchy. To disentangle these possibilities, we trained a set of control models in which we systematically permuted the assignment of the visual areas readouts to the last three layers (details in Appendix A.4). For example, in one such configuration, AL was assigned to layer 6, V1 to layer 7, and LM/RL to layer 8. To quantify the degree of alignment between readout assignment and the known cortical hierarchy (V1<LM $\lesssim$ RL<AL, Fig. 1A), we computed Kendall's $\tau$ between the readout layer indices and the anatomical order. Single-neuron prediction accuracy in AL increased

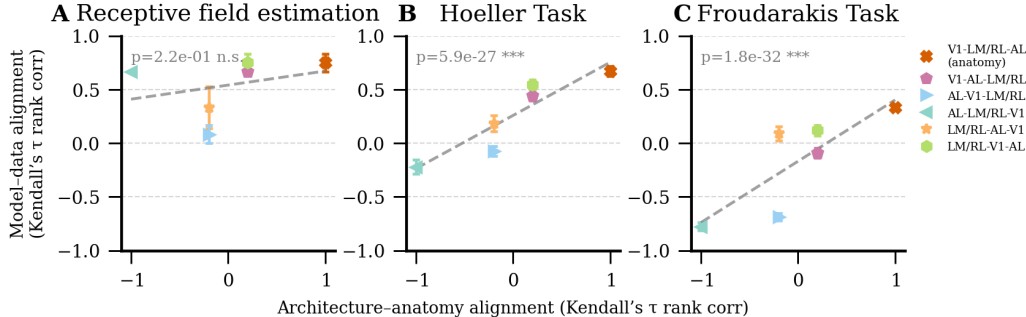

Figure 4: **Impact of architecture-anatomy alignment on model performance.** Agreement between various model-predicted and experimental observables orderings in different tasks, as a function of readout–hierarchy alignment, quantified by Kendall's $\tau$ between readout layer indices and the anatomical order (V1 < LM/RL < AL). Each dot corresponds to a different permutation of area-to-layer assignments across the last three layers of the network. Dashed lines indicate best linear fits. (**A**) RF size; (**B**) decoding accuracy (Hoeller task); (**C**) object discriminability (Froudarakis task).

systematically with the degree of alignment (Supp. Fig. S2), as well as in LM that recorded a minor positive trend. For V1 and RL, performance remained relatively stable across configurations. At the functional level, models with more anatomically aligned readouts exhibited stronger agreement with experimental data, especially in object recognition tasks (Fig. 4). These findings demonstrate that hierarchical readout is critical for capturing cortical functional organization, and that anatomical alignment of readout positions provides a powerful inductive bias for learning biologically plausible hierarchical representations in digital twin models.

**Alternative configurations of RL and LM readout depth.** To test whether our results depend on the assumption that LM and RL share the same readout depth, we trained two additional model variants. First, we developed a 9-layer model in which each visual area was assigned to a distinct depth, following the anatomical order V1 < LM < RL < AL. In this design, V1 read out from layer 6, LM from layer 7, RL from layer 8, and AL from layer 9. This variant produced similar overall results, maintaining consistent single-neuron accuracy and a hierarchical structure of receptive fields and object representations (Supp. Fig. S3). Second, motivated by evidence for dorsal/ventral stream separation in mouse visual cortex (Harris et al., 2019; D'Souza et al., 2022), we implemented a two-stream architecture. After layer 6, the network split into a ventral branch with a dedicated LM readout at layer 7, and a dorsal branch with RL and AL readouts at layers 7 and 8, respectively. This variant yielded similar prediction accuracy and maintained the global ordering of inter-area relationships (Supp. Fig. S3). Together, these results confirm that our main findings are robust to alternative architectural choices for LM and RL readout depth, and that grouping them in the main model is a reasonable simplification.

## 6 Discussion and limitations

Our work demonstrates that incorporating a biologically inspired hierarchical readout significantly enhances the ability of digital twins to replicate population-level phenomena in the visual cortex, particularly the emergence of invariant object representations. While a sufficiently large model, given enough training data, might achieve similar performance without a hierarchical readout, our architecture accomplishes this more efficiently in terms of both data and parameters. These findings contribute to a growing body of evidence that anatomical organization serves as a powerful inductive bias for neural network models of the brain (Kubilius et al., 2019; Margalit et al., 2024).

In our architecture, responses in each visual area are computed from a shared convolutional core, with parallel linear readouts from different layers. The core is hierarchical and cascaded, with deeper layers (e.g., layer 8 for AL) building on earlier ones (e.g., layer 7 for LM/RL). We do not, however, explicitly model inter-area communication as in biological circuits, where the output of one area informs the next. This simplification reflects current data-driven practices: routing activity from one readout to another might be more realistic but is unlikely to substantially change our results, as the shared core already approximates such computations. Moreover, because only a subset of

neurons is observed in each area, constraining downstream areas to rely solely on recorded upstream activity could limit performance. Allowing readouts to access shared internal representations instead yields a more complete embedding and improves generalization. Future work could explore modular architectures with explicit inter-area pathways to better capture distributed processing.

Hoeller et al. (2024) showed that invariant object recognition in mouse visual cortex and deeper layers of AlexNet relies on different underlying representations. In both biological and artificial networks, object representations become increasingly invariant to rotation along the hierarchy. However, the mouse visual system also preserves equivariant representations, where geometric transformations in the visual input lead to corresponding transformations of neuronal representations, whereas AlexNet appears to achieve invariance by suppressing these equivariant responses. Other studies have further shown that invariance and equivariance reflect not only network architecture but also training objectives and data, with task-driven models exhibiting mirror-symmetric viewpoint tuning (Cheon et al., 2022) and even randomly initialized networks displaying invariance properties (Farzmahdi et al., 2024). Our hierarchical model reproduces the experimentally observed increase in rotational invariance, but it remains unclear whether it also preserves equivariance. Addressing this question requires a layer-wise analysis of representational geometry and remains an important direction for future work.

Our findings also raise questions about the role of area RL in visual processing. Anatomically, RL ranks high in the cortical hierarchy–just above LM and below AL–and performs well in the object recognition task of Hoeller et al. (2024), yet it ranks lowest in the task of Froudarakis et al. (2020). This discrepancy may reflect differences in experimental conditions, such as the range of transformations or temporal structure. However, our model, which should be able to predict neural responses across conditions, predicts similar performance for RL across both tasks and hence fails to reproduce the result of Froudarakis. This suggests that task-level differences alone may not explain the experimental discrepancy. One possibility is that RL supports a distinct processing stream that is selectively engaged depending on task demands. Interestingly, anatomical and physiological studies have reported evidence for two partially segregated pathways in mouse visual cortex separating LM and RL (Wang et al., 2012; D'Souza et al., 2022). We tested this hypothesis in our dual-stream variant, which produced similar RL behavior, indicating that anatomical separation alone is insufficient. Instead, the divergent functional roles of RL may depend on additional factors not captured by our current framework, such as feedback from higher-order regions, behavioral state, or temporal integration across stimuli.

To evaluate representational hierarchy, we focused on two established metrics: receptive field size at the single-neuron level and invariant object recognition at the population level. While these are widely used indicators of cortical hierarchy, they capture only part of the full picture. Electrophysiological studies have revealed additional systematic variations across areas, such as differences in spatial and temporal frequency tuning (Glickfeld & Olsen, 2017), which future work could incorporate. At the population level, further studies might examine how cortical circuits support other computations beyond object recognition, such as spatial integration or motion processing. Although experimental data for these metrics remain limited, models like ours can help generate hypotheses and guide future experimental design.

Despite the incorporation of biologically inspired constraints, our model omits several important features of the visual system. It is entirely feedforward, lacking recurrence or feedback mechanisms that are known to shape neural responses, particularly in higher-level visual areas. Additionally, the model bypasses early sensory processing stages, such as those performed by the retina and lateral geniculate nucleus. Addressing these limitations will be crucial for closing the gap between digital twins and real cortical circuits, further refining both models and experimental approaches.

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

# Supplementary information

## A    Model Architecture, Training, and Evaluation

All code used for data processing, model training, and analysis is publicly available at `https://github.com/NeuroBLab/anatomically_inspired_digital_twin`.

### A.1    Baseline model architecture

The model architecture consists of four primary components: a core component, a behavioral module, a readout component, and a shifter module. The model takes videos and behavioral variables (pupil size, pupil center coordinates, running speed) as inputs. Videos are processed by the core component, returning an embedded representation of the visual stimuli. Behavior is transformed by the behavioral module into a latent representation which is stacked to the output of the last layer of the core. This representation is passed to the "readout", which determines the spatial position for each neuron and linearly combines the representation in the learned position with a set of learnable feature weights. The spatial position is further adjusted by the "shifter" module, which takes the pupil center's coordinates as input. Predicted responses are then generated frame by frame by applying a final nonlinearity (ELU+1) to maintain non-negativity. More details about each element are provided below.

**Inputs.** The inputs to the neural network model are:

1. Videos (Tensor V): A 5-dimensional tensor with shape (Batch, Channels, Frames, Height, Width), with Channels=1 (gray-screen movies), Height=36 and Width =64 pixels, respectively.

2. Behavior (Tensor B): a 3-dimensional tensor with shape (Batch, Channels, Frames) where Channels=4 represent pupil size, pupil center coordinates and running speed.

3. Pupil center (Tensor P): A 3-dimensional tensor with shape (Batch, Frames, Coordinates), where Coordinates=2 are the x, y positions of pupil center.

The input resolution (36×64 pixels) represents a trade-off between biological plausibility and computational efficiency, aligning with the limited visual acuity of mice and supported by prior work showing that lower-resolution training can improve predictivity (Nayebi et al., 2023). Performance remains comparable to higher-resolution models trained on the same dataset (Wang et al., 2025).

**Core.** The core component, inspired from (Höfling et al., 2022), is composed of 4 layers of 3D Convolutional Network with 16, 32, 64, and 128 channels, respectively. The 3D convolutions are factorized into spatial (2D) and temporal (1D) dimensions. Batch normalization is applied between the two convolutions. Spatial kernels are $11 \times 11$ in the first layer and $5 \times 5$ in subsequent layers, while temporal kernels are $11 \times 1$ in the first layer and $5 \times 1$ thereafter. Padding was added to both 2D and 1D convolutions to keep the original spatial dimensions unchanged and to maintain temporal causality, respectively. Batch normalization and an ELU + 1 non-linearity are applied after each layer. This component is shared across all recording sessions (see Appendix A.2).

This baseline design follows the Sensorium 2023 architecture (Turishcheva et al., 2024b), chosen for its suitability to dynamic visual stimuli and its status as the strongest available benchmark for mouse visual cortex. Earlier models used shallower CNNs with fewer channels (Lurz et al., 2020; Sinz et al., 2018; Klindt et al., 2017), making this architecture a natural starting point. We later examine deeper variants (6- and 8-layer cores) to assess whether increased expressivity alone can capture hierarchical transformations (see Section A.4).

**Behavioral module.** The behavioral module, inspired by (Li et al., 2023),consists of a 2-layer (4-128-32 units) multilayer perceptron (MLP), with Tanh and Dropout applied after each layer. A 1D convolution with kernel size 5 is applied along the temporal dimension, with pooling applied to enforce causality. The representation generated (a 32 channels vector frame-by-frame) is then stacked to the output of the core component. This component is shared across all recording sessions (see Appendix A.2).

**Readout.** To compute single responses during stimulus presentations, we adopted the readout described by (Lurz et al., 2020). Each neuron is associated with a learnable spatial position $p_n \in [-1, 1]^2$, used for bilinear interpolation at the specified positions in the core's output. This feature vector is then linearly combined with a learnable feature weight $w_n \in \mathbb{R}^{160}$ (128 channels coming from the core and 32 channels coming from the behavioral module) and a bias term to predict neuron activity frame-by-frame. A final (ELU + 1) non-linearity ensures non-negative responses. Each recording session is paired with its corresponding readout module.

We adopted the Gaussian spatial readout (Lurz et al., 2020), which has proven effective in previous large-scale mouse modeling studies. The number of features per neuron is fixed by the dimensionality of the accessed core layer (128 in our baseline). Larger feature counts, as used in foundation models (Wang et al., 2025), offer only marginal gains, suggesting our choice is sufficient and not a critical parameter for the results.

**Shifter.** The shifter is a multilayer perceptron (MLP) with three layers (2-5-5-2 units) (Sinz et al., 2018). It maps the pupil center coordinates to a shift applied to the neuron positions $p_n$. A Tanh nonlinearity at each layer ensures the output stays within the range $\in [-1, 1]^2$. Each recording session is paired with its corresponding shifter module.

## A.2 Training

**Training dataset.** We trained the model on publicly available functional calcium imaging data from the MICrONS dataset (Consortium et al., 2021), which integrates large-scale, multimodal functional recordings with structural reconstructions of the mouse visual cortex. All acquisition details, stimulus protocols, and preprocessing steps are comprehensively described in the original MICrONS publication (Consortium et al., 2021). Here, we report the key aspects relevant for reproducing our results.

Functional recordings were obtained using two-photon random access mesoscopy, capturing the activity of approximately 75,000 excitatory neurons across four cortical areas: the primary visual cortex (VISp) and three higher-order visual areas (VISlm, VISrl, and VISal). The dataset consists of 14 recording sessions from a single transgenic mouse passively viewing visual stimuli. Visual stimuli consisted of diverse gray-scale natural and parametric videos, including cinematic movie clips, naturalistic sports videos, and rendered 3D scenes, designed to probe a broad range of neuronal tuning properties. Behavioral data were recorded simultaneously and aligned with neural activity. During imaging, the mouse was head-fixed but free to walk on a treadmill. Behavioral variables were precisely tracked and synchronized with neural recordings, including running speed measured via a rotary optical encoder (57–100 Hz) and eye movements recorded at 20 Hz using an infrared camera capturing pupil center coordinates and dilation.

We used all available sessions, which comprise 83,222 neurons from primary visual cortex (V1), 14,817 neurons from lateral medial area (LM), 12,599 from rostrolateral area (RL), and 4,734 from anterolateral area (AL). All videos were isotropically downsampled to a resolution of $36 \times 64$ pixels per frame. Functional and behavioral signals were resampled to 30 Hz by linear spline interpolation. Visual stimuli were normalized by subtracting the mean and dividing by the standard deviation across all recording sessions. Behavioral data, and pupil center coordinates were normalized by subtracting the mean and dividing by the standard deviation of each recording session. Neural responses were standardized to ensure non-negativity.

**Loss Function and Training Details.** Batches consisted of 150 consecutive frames, randomly sampled from 300-frame trials. Training used a batch size of 2. Learning rate was set to 0.005. AdamW (with default hyperparameters) was employed for parameter optimization, minimizing the negative Poisson log-likelihood loss:

$$\mathcal{L}_s^{Poisson} = \sum_{t=1}^{n_t} \sum_{i=1}^{n_s} r_{i,t} - o_{i,t}\log(r_{i,t})$$

between the recorded responses $o$ and predicted responses $r$, where $n_t$ is the number of frames in one batch and $n_m$ is the number of neurons for recording session $s$. Parameter updates were applied after completing one full pass through all sessions, a strategy previously found to enhance performance (Li et al., 2023). Training employed a learning rate schedule with linear warm-up during the first 10 epochs. After each epoch, we computed the correlation to average (see below) between predicted and

observed responses on a validation set averaged across all neurons (excluding the initial 50 frames). The validation set is composed by 6 videos which were repeated 10 times during each recording session and held-out during training. If this score did not improve for eight epochs, training was halted, and the model was reverted to the checkpoint with the highest validation score. The learning rate was then reduced by a factor of 0.3 unless the maximum number of decay steps ($n = 4$) had been reached, in which case training was terminated. Each model described was trained with three different random seeds.

### A.3 Evaluation Metrics

Model evaluation used two metrics: correlation to average and normalized correlation coefficient $CC_{\mathrm{norm}}$. The former is calculated as follows:

$$\text{avg. corr.}(r, o) = \frac{\sum_i (\bar{r}_i - \bar{r})(o_i - \bar{o})}{\sqrt{\sum_i (\bar{r}_i - \bar{r})^2 \sum_i (o_i - \bar{o})^2}} \tag{S1}$$

where $\bar{r}_i = \sum_{j=1}^{J} r_{i,j}$ is the average response across J repeats.
The latter is defined as (Schoppe et al., 2016)

$$CC_{\mathrm{norm}} = \frac{CC_{\mathrm{abs}}}{CC_{\mathrm{max}}}$$

where $CC_{\mathrm{abs}}$ and $CC_{\mathrm{max}}$ are defined as:

$$CC_{\mathrm{abs}} = \frac{\mathrm{Cov}(\bar{r}, \bar{o})}{\sqrt{\mathrm{Var}(\bar{r})\mathrm{Var}(\bar{o})}}, \quad CC_{\mathrm{max}} = \sqrt{\frac{N\mathrm{Var}(\bar{o}) - \overline{\mathrm{Var}(o)}}{(N-1)\mathrm{Var}(\bar{o})}}$$

Here, where $r$ is the predicted response and $o$ is the observed response to $N$ repeated stimuli; $\bar{o}$ and $\bar{r}$ are the average recorded and predicted responses across all trials. All plots showing single-neuron prediction performance report the median $CC_{\mathrm{norm}}$ across neurons for each session averaged over all the sessions, evaluated on the validation set.

#### A.3.1 Kendall's $\tau$ definition

Kendall's Tau is a rank-based statistic used to assess the ordinal relationship between two variables, specifically quantifying the degree of agreement or disagreement in their relative rankings. It is particularly useful when analyzing data with ordinal scales or when the assumption of linearity in Pearson's correlation is not valid. The statistic is defined as:

$$\tau = \frac{P - Q}{(P + Q + T)(P + Q + U)}$$

where:

- $P$ represents the number of concordant pairs, which are pairs of observations $(x_i, y_i)$ and $(x_j, y_j)$ where the relative order of both variables is the same, i.e., if $x_i < x_j$, then $y_i < y_j$ (or if $x_i > x_j$, then $y_i > y_j$).
- $Q$ represents the number of discordant pairs, where the relative order of the two variables is reversed, i.e., if $x_i < x_j$, then $y_i > y_j$, or if $x_i > x_j$, then $y_i < y_j$.
- $T$ is the number of tied pairs in the first variable,
- $U$ is the number of tied pairs in the second variable.

Kendall's Tau measures the balance between concordant and discordant pairs, with values ranging from $-1$ to $+1$. A value of $\tau = 1$ indicates perfect agreement between the two variables' rankings, while $\tau = -1$ indicates perfect disagreement. A value of $\tau = 0$ suggests no association between the variables' rankings.

### A.4 Model variations

To systematically investigate the influence of architectural choices on model performance, particularly concerning the representation of invariant object recognition across visual areas, we developed several model variations:

- **Area-balanced loss function.** Given the substantial imbalance in neuron counts across visual areas, standard training would bias the model toward optimizing V1 predictions at the potential expense of higher visual areas. We addressed this by implementing an area-normalized loss function where each area's contribution is scaled by the inverse of its neuron count. This weighting scheme ensures each visual area contributes equally to the loss function regardless of neuron count, preventing the model from optimizing predominantly for V1. This method resulted in a reduced loss magnitude which had to be balanced by an increase in the learning rate. We searched the optimal learning rate training models with a single session (5_6) to allow extensive experimentation.

- **All-layer readout model.** We explored a readout that instead of looking only at the output of the last layer of the core can read from representations extracted at different depths, possibly extending the model's capability to capture visual features at multiple levels of abstraction. In this variant, each neuron is assigned learnable feature weights that are applied to output channels from all core layers. In order to encourage sparsity across channels, we explored various configurations of L1 regularization (5, 10, 25, 50, 200) applied to the learned weights. To allow extensive search, the exploration was performed using a single recording session (5_6). We found this model variant prone to overfitting suggesting that unconstrained multi-layer access does not provide the appropriate inductive bias for modeling the visual hierarchy.

- **Deeper network architectures.** To investigate whether the hierarchical processing was limited by model expressivity, we developed variants of the baseline model with increased depth (6 and 8 layers). We extended the baseline 4-layer architecture with two or four additional convolutional layers (with 256 and 128 channels in the 6-layer case and 256, 512, 256 and 128 channels in the 8-layer case). These deeper architectures were trained with the same protocol as the baseline model, allowing us to assess whether increased depth alone could capture hierarchical transformations without explicit anatomical constraints.

  Varying the depth of the convolutional core affects the number of internal parameters, but the number of readout parameters was kept constant across all model variants (including both hierarchical and non-hierarchical readouts). This design choice allows us to isolate the effect of core depth without conflating it with decoder complexity. The distribution of parameters across model variants is shown in Table 1.

- **Anatomically-constrained hierarchical readout.** Our key proposed model implements a biologically inspired readout scheme that mirrors cortical anatomy. Each visual area is modeled as reading out from a specific depth in the network, reflecting their position in the anatomical hierarchy. We built an 8-layer model (since the 6-layer model showed the best performances with no significant improvements with deeper networks) where V1 neurons read out from the 6th layer, LM and RL from the 7th and AL from the last one. This constraint enforces inputs to be processed following the cortical anatomy, enabling each area to learn a tailored set of filters with more freedom and to benefit from representations generated for lower-level areas. To ensure a fair comparison with other models, each area still read from an output consisting of 128 channels (resulting in a 16-32-64-128-256-128-128-128 channels for the corresponding layers).

- **Control hierarchical readouts.** To verify that performance improvements from our hierarchical model derive from meaningful anatomical constraints rather than merely from the assignment of areas to different layers, we implemented the following control: for each block (V1, LM/RL, AL) we permuted the order with which they were assigned to the last three layers (for example, AL being assigned to the 6th layer, V1 to the 7th layer and LM/RL to the 8th layer). These control models help distinguish between benefits arising from anatomically meaningful constraints versus those stemming merely from the flexibility of assigning different areas to different layers.

Table 1: **Parameter counts across model variants.** Core parameters vary with depth, while readout parameters are held constant.

| Model Variant | Core Parameters | Readout Parameters |
|---|---|---|
| Baseline (4 layers) | 382,512 | 17,396,557 |
| Baseline (6 layers) | 2,432,816 | 17,396,557 |
| Baseline (8 layers) | 10,629,424 | 17,396,557 |
| Hierarchical (8 layers) | 3,417,392 | 17,396,557 |
| Wang et al. (2025) | 4,528,640 | 54,377,262 |

# B    Repetition of experiments to probe functional hierarchy

## B.1    Receptive field extraction and analysis

**Estimation of artificial receptive fields.** We estimated artificial receptive fields (RFs) of units in trained models following the procedure introduced by Li et al. (2023). Each trained model was presented with $N = 500{,}000$ white noise images sampled from a uniform distribution. The receptive field $a_{\mathrm{RF},i}$ of unit $i$ was computed as the weighted sum of all input images, with weights given by the unit's response to each image:

$$a_{\mathrm{RF},i} = \sum_{n=1}^{N} F(x_n)_i \cdot x_n, \quad x_n \sim \mathcal{U}(1 \times 36 \times 64),$$

where $F(x_n)_i$ denotes the activation of unit $i$ in response to input $x_n$, and $\mathcal{U}(1 \times 36 \times 64)$ indicates sampling from a uniform distribution over the input shape.

**Gaussian fitting and receptive field size.** To characterize spatial properties of each receptive field, we fitted a 2D Gaussian function to extract its center location (mean) and spatial spread (covariance). Prior to fitting, each receptive field map was smoothed using a Gaussian filter, mean-centered, and transformed by taking its absolute value to improve fitting stability. Gaussian fitting was performed using SciPy's `curve_fit()` function. The receptive field size was defined as $\pi * \sigma_x * \sigma_y$ (where $\sigma_x$ and $\sigma_y$ are the standard deviations along the two principal axes of the fitted Gaussian).

**Conversion from pixels to degrees of visual angle.** To convert receptive field sizes from pixels to degrees of visual angle, we applied a conversion factor calculated based on the spatial resolution and viewing conditions of the input stimuli given in (Consortium et al., 2021). Visual stimuli were presented to the left eye on a 31.8x56.5cm (heightxwidth) monitor with a resolution of 1,080x1,920 pixels, positioned 15cm away from the eye. Given this setup, each pixel corresponds to approximately 0.88 cm on the screen. Assuming the mouse fixates on the center of the screen and the eye is perpendicular to the screen, the visual angle subtended by one pixel is calculated as:

$$\theta = \arcsin\left(\frac{0.88}{\sqrt{0.88^2 + 15^2}}\right) \approx 0.0586 \text{ radians} \approx 3.36°.$$

Because receptive field size is computed as the product of standard deviations along two axes (i.e., an area), the size in degrees squared is obtained by multiplying the pixel-based area by $(3.36°)^2$.

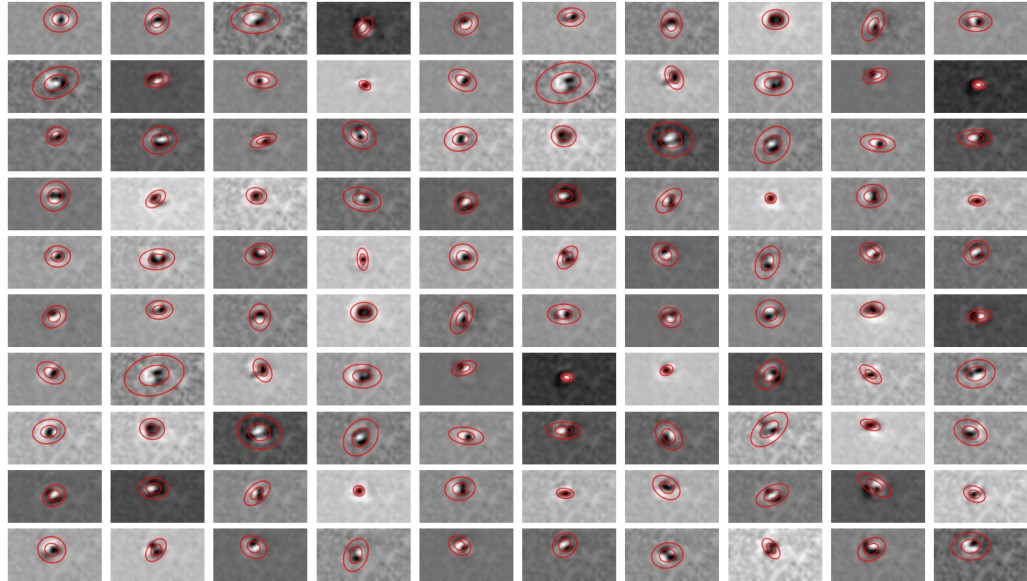

Figure S1: **Example receptive fields estimated by the model.** Representative receptive fields obtained by aggregating white noise responses of individual units. Red contours indicate iso-response levels of the best-fitting 2D Gaussian at one and two standard deviations. These fits were used to estimate receptive field centers and sizes.

**Selection of units for analysis** First of all units were filtered based on their modeling accuracy: neurons with a correlation to average below 0.35 were discarded to avoid producing model's artifacts. Furthermore, not all receptive fields yielded reliable Gaussian fits. We therefore excluded units based on two additional criteria: (1) the center of the fitted Gaussian lay outside the stimulus image boundaries, indicating poor localization; (2) either $\sigma_x > 16$pixels or $\sigma_y > 14$pixels, which filtered out outliers with implausibly large RFs. This selection ensured inclusion of units with well-defined and biologically plausible receptive fields. Figure S1 presents representative examples of accepted receptive fields.

**Summary statistics and visualization** For each visual area modeled, the average receptive field size was computed across all accepted units and used for subsequent analyses.

## B.2 Hoeller task replication

To evaluate the capacity of our digital twin model to capture hierarchical improvements in rotation-invariant object recognition, we replicated the object classification task of Hoeller et al. (2024) using simulated neural responses generated by our model. While our analysis pipeline followed the structure described by Hoeller et al., it was implemented independently.

**Stimuli.** We constructed a stimulus set analogous to that used by Hoeller et al. It comprised 92 static, grayscale images representing 92 distinct object classes, selected from copyright-free sources using the COCO dataset classes as a reference. Images underwent preprocessing involving masking with a circular aperture and histogram equalization. Each image was then rotated in-plane to create 15 versions, spanning 360deg in uniform increments of $\Delta\phi = 24$deg. For presentation to the digital twin, which operates on video inputs sampled at 30 fps, each static rotated image was embedded within a short video clip. Each clip consisted of an initial 50 frames ($\sim 1.67s$) of a uniform gray screen, followed by 15 frames ($\sim 0.5s$) where the static rotated image was displayed. This structure allowed for the analysis of responses specifically during the image presentation period, distinct from any preceding baseline or transient activity. Input stimuli were appropriately normalized based on statistics derived from the dataset used for training the digital twin.

**Model responses.** Digital twins trained on the MICrONS dataset were used to predict responses the rotated object video clips. For each stimulus presentation (a specific object identity at a particular rotation angle), the model processed the corresponding 65-frame video clip. The resulting simulated neural activity trace for each neuron was then averaged over the final 15 frames, corresponding to the period of static image presentation only. This produced a single scalar value representing the response magnitude of each neuron to that specific stimulus. Responses were generated for simulated neurons located in modeled visual cortical areas V1, LM, AL, and RL.

**Reliable neuron selection.** From the pool of simulated neurons in each visual area (V1, LM, AL, RL), we selected those exhibiting reliable responses. Reliability was defined based on the neuron's performance during the original model training phase: we included neurons whose predicted activity on a held-out validation set showed a correlation coefficient greater than 0.35 with the trial-averaged predicted response for that neuron.

**Preprocessing.** For each selected neuron, the mean simulated response recorded during the initial 50 gray-screen frames was calculated and subtracted from the responses during the image presentation frames for baseline correction. Following this, the responses were standardized. Using only the data corresponding to stimuli designated as training classes (defined per iteration, see below), the standard deviation of responses for each neuron was computed and used to z-score all responses (from both training and test classes). Finally, the mean response across these training classes was subtracted from all responses to mean-center the data relative to the training set.

**Decoding analysis.** The preprocessed, high-dimensional population response vectors for each visual area were projected onto a lower-dimensional subspace using Principal Component Analysis (PCA). The PCA was fitted using only the training class data, and the top 100 principal components, capturing the directions of highest variance, were retained for subsequent classification. A linear Support Vector Machine (SVM) classifier was trained to distinguish between the 92 object classes based on the 100-dimensional PCA representations. We used the implementation 'LinearSVC' from the scikit-learn library, configured with a regularization parameter C=1 and number of iterations equal to 1000 to ensure convergence. To evaluate the classifier's ability to generalize to unseen rotations, we employed a 15-fold cyclic cross-validation procedure based on the rotation angle, identical to that used by Hoeller et al. (2024). In each fold, the data corresponding to three consecutive rotation angles were held out. The linear SVM was trained using the model responses associated with the remaining 12 rotation angles, encompassing stimuli from all 92 object classes.

**Performance and repetitions.** The classifier's performance was then evaluated on its ability to predict the identity of the stimuli belonging to the 2 designated test classes (see below) presented at the middle rotation angle of the three held-out angles. Classification performance for each fold was measured as the accuracy (fraction of correctly classified test stimuli). The overall accuracy for one iteration of the analysis was the average accuracy across the 15 folds. Since the performance depends on which 2 out of the 92 classes are chosen as test classes, the entire analysis procedure—including the random selection of 2 test classes, subsequent preprocessing, PCA fitting, SVM training, and 15-fold cross-validation testing—was repeated 50 times. Each repetition used a different random seed to ensure different pairs of test classes were selected from the 92 classes. The final reported accuracy for each visual area represents the mean accuracy ± standard error of the mean (SEM) calculated across the six MICrONS sessions and three training seed for each model.

### B.3  Froudarakis task replication

To assess the digital twin's ability to support object identity discrimination based on dynamic visual input, mirroring the experimental logic of Froudarakis et al., we replicated their paradigm using simulated neural responses generated by our model in response to videos of objects undergoing continuous identity-preserving transformations (rotation, translation, scale, illumination condition). Our stimuli and analysis pipeline were designed based on the description provided by Froudarakis et al. but implemented independently.

**Stimuli.** We generated a stimulus set comprising videos of four distinct three-dimensional objects using 3D rendering in Blender, aiming to closely resemble those used in the original study by Froudarakis et al.. Each object identity was subjected to continuous, identity-preserving transformations involving random variations in position (translation along X, Y, Z axes), scaling (size variation), rotation (tilt), and environmental lighting conditions (intensity and position). These transformations

resulted in smooth, coherent object motion against a uniform gray background. For each of the 4 object identities, we rendered 500 unique video clips. Each clip was 10 seconds in duration, rendered at 30 frames per second (fps), resulting in 300 frames per clip. Input video stimuli were normalized based on statistics derived from the dataset used for training the digital twin, consistent with the procedure used for the Hoeller et al replication described above.

**Model responses.** We employed the same pre-trained digital twin model used in the previous experiment to predict neural responses. The model processed each 10-second (300-frame) video clip. To account for potential neural adaptation or onset transients, we excluded the simulated responses corresponding to the initial 30 frames ( 1 second) of each clip from the analysis. The subsequent 270 frames of activity were then divided into 18 contiguous, non-overlapping temporal bins, each spanning 15 frames (500 ms). The simulated neural activity trace for each neuron was averaged within each 500 ms bin. This procedure yielded a sequence of 18 response vectors per video clip for each simulated neuron, where each vector represented the average activity level in a successive 500 ms interval. Responses were generated for simulated neurons located in the modeled visual cortical areas.

**Reliable neuron selection.** For each visual area (V1, LM, AL, RL), we first identified a pool of reliable simulated neurons. Reliability was assessed based on performance during the model's training phase: neurons were included if the correlation coefficient between their trial-averaged predicted response and the corresponding ground truth neural activity exceeded 0.35 on a held-out validation dataset.

**Decoding analysis.** Qualifying neurons were pooled together for each visual area. Prior to classification, the neural response vectors underwent preprocessing steps within each fold of the cross-validation procedure described below. First, using only the data designated for training within a given fold, the mean and standard deviation of each feature (i.e., each neuron's response) were calculated. All features in both the training and testing sets of that fold were then standardized by subtracting the training set mean and dividing by the training set standard deviation (z-scoring).

For each repetition of the decoding procedure, a subset of 128 neurons was randomly selected without replacement from the pool of reliable neurons available for the target visual area, consistent with the population size sampled by Froudarakis et al. To classify the object identity based on the 128 selected neural responses, we employed a multi-class logistic regression approach. Specifically, a set of binary logistic regression classifiers were trained using a one-vs-rest strategy, where each classifier learned to distinguish one object class from the remaining three. These classifiers were optimized using an iterative numerical method to maximize the likelihood of the observed data under the logistic model. The optimization procedure did not include a regularization penalty term on the classifier weights, but did fit an intercept term for each classifier.

A 10-fold stratified cross-validation scheme was used to evaluate the generalization performance of the classifiers. The full dataset of 500 ms response vectors (pooled across all clips and objects) was randomly partitioned into 10 equally sized folds. Stratification ensured that the relative proportion of samples belonging to each of the four object classes was maintained within each fold. For each iteration of the cross-validation, one fold served as the test set, while the remaining nine folds were used for training the logistic regression classifiers (including the calculation of standardization parameters). This process was repeated 10 times, with each fold serving as the test set exactly once.

**Performance metric.** The primary measure for object discriminability was the Mutual Information ($MI$), quantified in bits, between the true object labels ($c$) and the labels predicted by the classifier ($\hat{c}$) on the held-out test sets across the 10 folds. $MI$ was calculated from the aggregated confusion matrix $C$ (summed across folds) using the formula:

$$\text{MI}(c, \hat{c}) = \sum_{i=1}^{4} \sum_{j=1}^{4} p(c_i, \hat{c}_j) \log_2 \left( \frac{p(c_i, \hat{c}_j)}{p(c_i)p(\hat{c}_j)} \right)$$

where $p(c_i, \hat{c}_j)$ represents the joint probability of true class $i$ and predicted class $j$ (derived from the normalized confusion matrix), and $p(c_i)$ and $p(\hat{c}_j)$ are the marginal probabilities (derived from the row and column sums of the normalized confusion matrix, respectively).

**Repetitions and reporting.** To ensure robust estimation of performance, the entire analysis procedure—encompassing the random subsampling of 128 neurons and the subsequent 10-fold stratified cross-validation with MI calculation—was repeated 50 times for each visual area. Each repetition

utilized a distinct random seed to ensure different neuron subsets and data partitions were used. The final reported discriminability value for each visual area is the mean $MI$ computed across these 50 independent repetitions, presented as mean ± standard error of the mean (SEM) calculated across the six MICrONS sessions and three training seed for each model.

# C  Implementation and implications

## C.1  Computational resources

All experiments were run on two different infrastructures based on their computational demands. Each model (including baseline and hierarchical variants) was trained on a single NVIDIA A100 80GB PCIe GPU, paired with a dual-socket Intel Xeon Gold 5317 CPU (24 cores total). Training took approximately 3.8 days per model to reach convergence. Memory usage during training remained well within the 80GB VRAM limit. The receptive field estimation analysis was conducted using the same hardware as model training (A100 GPU and Xeon CPU) and required approximately 12 hours per session with settings described above. Both the Hoeller and Froudarakis task replications were performed on a machine with an NVIDIA RTX A4000 (16GB) GPU and a single-socket Intel Xeon Gold 6248R CPU (48 threads) with 128GM RAM associated. Each run took approximately 2.5 hours, and most of the computation was CPU-bound. All model trainings were repeated three times with different random seeds and all experimental tasks were repeated for each trained model. Execution was parallelized where feasible.

## C.2  Broader impact

Our work represents a step toward developing more biologically informed digital twin models, enabling the study of invariant object recognition in silico. As these models improve, they can provide a powerful framework for conducting experiments that are not feasible in animal studies and for reducing the need for invasive animal experiments. Eventually, as these models advance and our understanding of brain function deepens, they may become valuable tools for investigating the neural basis of neurological disorders and exploring potential treatments and interventions.

# D    Additional supplementary figures

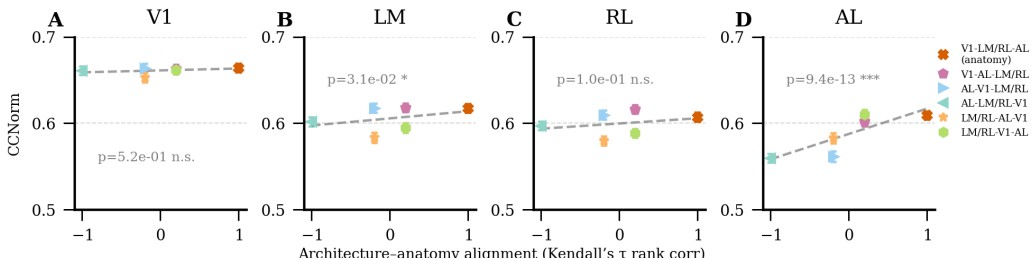

Figure S2: **Prediction performance across areas as a function of readout–hierarchy alignment.**
Single-neuron prediction performance in each area plotted against the alignment between readout
layer indices and anatomical area order (V1 < LM/RL < AL), quantified by Kendall's $\tau$. Each dot
represents a model defined by a distinct permutation of area-to-layer assignments across the last three
layers of the network (as in all models of Fig. 4). Dashed lines indicate best linear fits.

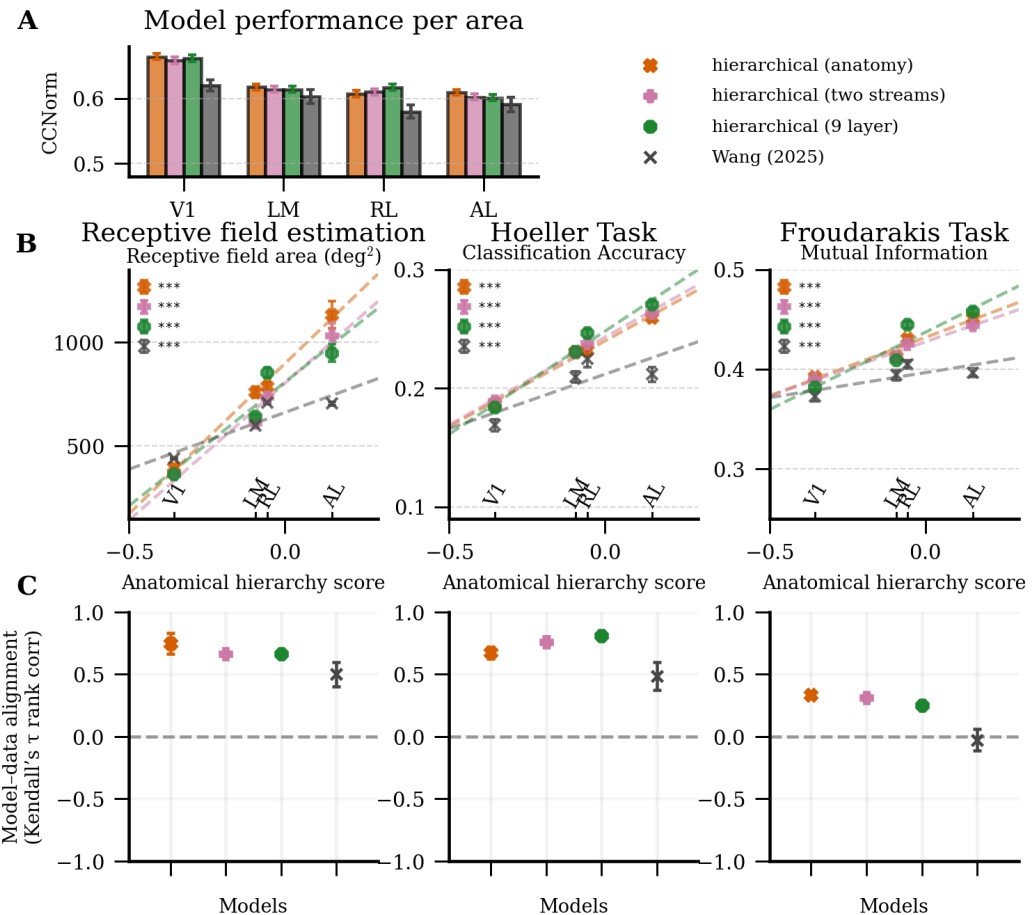

Figure S3: **Comparison of alternative large-scale and hierarchical model architectures.** (**A**) Single-neuron prediction performance across areas for the four tested architectures: the main hierarchical model (8 layers), the 9-layer variant with distinct readout depths for each area, the two-stream variant separating ventral (LM) and dorsal (RL–AL) branches, and the large-scale foundation model of Wang et al. (2025). Note that the performance of the Wang et al. (2025) model may be influenced by how we entered behavioral variables into its publicly available implementation, which we used directly. Importantly, this does not affect the conclusions of the following analyses, as all other in-silico experiments were conducted without behavioral inputs. (**B**, left to right) Receptive-field size, classification accuracy in the Hoeller task, and object discriminability in the Froudarakis task, each plotted as a function of anatomical hierarchy score (dots) with best linear fits (dashed lines). (**C**) Agreement between model-predicted and experimentally observed orderings in (B), quantified by Kendall's $\tau$ rank-correlation coefficient. All alternative hierarchical variants reproduced the gradient in invariance and maintained consistent single-neuron accuracy across areas, whereas the Wang et al. (2025) model failed to recover the fine-scale hierarchical distinctions among higher-order regions.

