# OpenReview forum: "Anatomically inspired digital twins capture hierarchical object representations in visual cortex"
_NeurIPS.cc/2025/Conference — NeurIPS 2025 poster_

### Official Review · Reviewer_vF7y · 2025-06-25

**Clarity:** 3
**Significance:** 4
**Originality:** 3
**Rating:** 5
**Confidence:** 3

**Summary:**

The paper presents a novel way of training digital twin models of mouse visual cortex. Opposed to established approaches, which often only optimize the prediction performance in the final layer, the paper proposes a 'biologiclally inspired readout' procedure, where prediction performance is optimized along different levels of the network hierarchy in comparison to corresponding neural data. It is demonstrated that training of digital twin models using the novel method improves significantly the reproduction of the variations of different properties along the visual hierarchy, including receptive field size, decoding accuracy, and object discriminability in comparison to special data sets recorded in mouse visual cortex.

**Questions:**

On major question that might be discussed / reviewed is why people previously strongly defended the idea of an end to end training of neural response reproductions by neural models. Given that these structures include strong redundancies, it seems potentially not surprising that end-to end training to reproduce V1 responses with a deep hierarchical  model should provide a good reproduction of the variation of properties long the visual hierarchy in the cortex. In this sense, the author's approach seems much more reasonable. It would be interesting to discuss the arguments why this was done differently before.

Small points:

ELU+1:  maybe explain briefly

Supplement A2: down-sampling to 36 x 64 pixels of the video in the first place seems to be very coarse in terms of spatial resolution;
why does this make sense, or does not constrain the obtained results?

**Ethical Concerns:**

["NO or VERY MINOR ethics concerns only"]

**Final Justification:**

I think technically this is a very solid paper. While still I think that the theoretical interpretation might stimulate some debatable points, however it will make a very useful contribution to the literature. The concerns of the other reviewers, e.g. about details of the biological plausibility, seem have to be addressed largely in a satisfying way. Therefore, I maintain my score, even though I am not an expert for models of the mouse brain.

**Limitations:**

yes

**Quality:**

4

**Strengths And Weaknesses:**

The paper is very well written, well structured, and the developed approach is well justified. Variations of the proposed methods are tested, and by  appropriate model comparisons and ablation studies it is convincingly demonstrated that the proposed novel training approach really results in better reproduction of invariance properties of neurons along the visual hierarchy than neural models trained with the classical end-to-end approach.

A little weakness in my view is the failure to quite critical literature, e.g. about hierarchies in visual cortex, before 2011.

In addition, some assumptions of the model could be more clearly motivated in the text, e.g. why the chosen number of layers, neurons etc of the core unit. How are the the chosen readout model parameters chosen, and in how far are they critical , or not, for the results.

---

> ### Author Rebuttal · Authors · 2025-07-30
>
> We thank the reviewer for their constructive and encouraging feedback. We are pleased that the biological motivation, structure, and empirical support of our approach were positively received. We appreciate the support for our new approach to modeling neural responses. Below, we address the specific points raised in the review.
>
> ---
>
> **1. Contextualizing our approach in prior literature**
> Previous studies have explored various strategies to understand the visual cortex, including linear-nonlinear (LN) models (Heeger, 1992; Simoncelli et al., 2004), energy models (Adelson & Bergen, 1985), and others. Deep learning has significantly advanced predictive performance, especially through convolutional neural networks (CNNs) trained on image classification tasks (e.g., Nayebi, 2023) or directly on neural data (e.g., Klindt et al., 2017; Wang et al., 2023). For a comprehensive review, see [1]. Importantly, many prior models incorporate biological constraints. For example, Antolík et al. (2016) proposed a structured model of V1 responses inspired by the feedforward visual hierarchy, where populations of V1 neurons share thalamic inputs and are organized into simple and complex-like layers. While such models have been surpassed in performance by end-to-end CNNs, the latter are often still motivated by biological principles. A key innovation, introduced by Klindt et al. [2] and adopted in our work, is the factorization of feature and spatial position into orthogonal components. Traditional approaches often treat each neuron independently, but the visual system exhibits spatial equivariance: similar computations recur across nearby locations. Leveraging this shared structure enables models to learn a common nonlinear feature space, leading to more powerful and generalizable predictions than those based on isolated neurons.
> Our introduction of a hierarchical readout continues this biologically inspired modeling tradition. We first recognized that standard models, with a shared readout from the final feature layer, perform well at predicting individual neuron responses, but fail to capture population-level response structure. This motivated us to introduce a hierarchical readout variant, which explicitly models population organization by allowing structured variation in how different neurons access shared features. This approach enables the model to capture higher-order response patterns across the population, going beyond traditional single-neuron metrics.
>
> [1] Turishcheva et al. “The Dynamic Sensorium competition for predicting large-scale mouse visual cortex activity from videos.” arXiv (2023)
> [2] Klindt et al. “Neural system identification for large populations separating ‘what’ and ‘where’.” Advances in Neural Information Processing Systems (2017)
>
> ---
>
> **2. On prior literature about cortical hierarchies**
> Regarding the concern about missing citations to earlier literature on cortical hierarchies, we agree this is an important point. For mice, we will add earlier references [1,2] that demonstrate prior knowledge of visual areas beyond V1. For the anatomical hierarchy specifically, [3] was, to our knowledge, the first to systematically characterize interareal connectivity and propose a hierarchical structure based on anatomical data. Please let us know if there are additional references we may have missed. While the literature on primates is indeed more extensive and older, our study focuses on the mouse visual system; for space reasons, we chose to cite a review article [4] to provide broader context.
>
> [1] Wang & Burkhalter. “Area map of mouse visual cortex.” J. Comp. Neurol. (2007)
> [2] Marshel et al. “Functional specialization of seven mouse visual cortical areas.” Neuron (2011)
> [3] Wang et al. “Network analysis of corticocortical connections reveals ventral and dorsal processing streams in mouse visual cortex.” J. Neurosci. (2012)
> [4] DiCarlo et al. “How does the brain solve visual object recognition?” Neuron (2012)
>
> ---
>
> **3. Motivation and impact of architectural choices**
> For the core, we built upon the architecture introduced in the Sensorium 2023 challenge ([1]), which employed a four-layer factorized 3D convolutional network with increasing channel depth (16, 32, 64, 128). This choice was motivated by its suitability for processing dynamic stimuli with spatial and temporal resolution comparable to those used in our study. While previous models in the literature have generally used shallower cores, e.g., [2] employed a four-layer 2D CNN with 64 channels per layer; [3] and [4] used three layers with up to 48 channels, the Sensorium architecture represented the best publicly available option for mouse visual modeling and served as a practical benchmark.
> Importantly, our work is among the first to systematically test how increasing core depth affects neural prediction performance. We extended the core from 4 to 8 layers and observed a marked improvement (8.4%) in single-neuron prediction accuracy. However, this modification alone did not enhance alignment with population-level hierarchical organization. This result suggests that deeper networks improve single-neuron fitting but do not resolve functional mapping across cortical areas, highlighting the necessity of architectural biases like our anatomically-inspired hierarchical readout.
> For the readout, we adopted the Gaussian spatial readout proposed by [2], which currently represents the best-performing readout framework for mouse visual cortex. The dimensionality of the features used in the readout is determined directly by the channel count of the core layer being accessed, e.g., 128 features per neuron from the final layer in our case. We did not extensively test this dimensionality, but it appeared sufficient for capturing the target responses. As a point of comparison, [5] used 512 features per neuron, yet reported only marginal gains in prediction accuracy, suggesting limited sensitivity to readout dimensionality beyond a certain threshold.
> We will revise the manuscript to explicitly include this rationale and emphasize that our work systematically explores design choices that have previously been adopted without extensive justification. This includes evaluating how architectural decisions, such as core depth and readout configuration, affect both single-neuron accuracy and population-level representational structure.
>
> [1] Turishcheva et al. “The Dynamic Sensorium competition for predicting large-scale mouse visual cortex activity from videos.” arXiv (2023)
> [2] Lurz et al. “Generalization in data-driven models of primary visual cortex.” ICLR (2021)
> [3] Sinz et al. "Stimulus domain transfer in recurrent models for large scale cortical population prediction on video." NeurIPS (2018)
> [4] Klindt et al. “Neural system identification for large populations separating “what” and “where”.” NeurIPS (2017)
> [5] Wang et al. “Foundation model of neural activity predicts response to new stimulus types.” Nature (2025)
>
> ---
>
> **4. Clarifications on minor points**
> We will briefly clarify the use of the ELU+1 activation function in the main text. This ensures strictly positive activations, which is more biologically realistic given that firing rates are inherently non-negative.
> Regarding the input video resolution (36 × 64 pixels), this reflects a trade-off between biological plausibility and computational efficiency. The low resolution aligns with the limited visual acuity of mice and is further supported by evidence that lower-resolution training can improve neural response predictivity (e.g., Fig. 3 in [1]). Notably, our model achieves single-neuron prediction performance comparable to that reported in [2], which also modeled responses from the MICrONS dataset but used higher resolution (144×256 pixels), enabled by extensive pretraining across multiple mice. This comparison supports the conclusion that lower resolution is sufficient for accurate modeling. We will clarify this rationale in Supplement A2.
>
> [1] Nayebi et al. “Mouse visual cortex as a limited resource system that self-learns an ecologically-general representation.” PLOS Comp Bio (2023)
> [2] Wang et al. “Foundation model of neural activity predicts response to new stimulus types.” Nature (2025)
>
> ---
>
> Once again, we thank the reviewer for their valuable feedback and are confident that the proposed revisions will strengthen the manuscript.

---

> ### Comment · Reviewer_vF7y · 2025-08-04
> **Point about possible theoretical implications**
>
> Many thanks to the authors for their comprehensive answers to most of my comments.
>
> I still have one theoretical question that is not completely answered by the authors' rebuttal.
> From the viewpoint of NN-based learning it seems rather expected that an architecture with
> sufficient complexity that is trained end-to-end can provide good fits for neural data,
> while not necessarily providing good matching of the behavior of neural units at
> intermediate levels of the hierarchy. That this matching becomes better by constraining
> the network by additional data that constrains also the behavior of units at
> intermediate levels of the model seems not very surprising. Apparently, in this field people so far chose to focus on end-to-end training, not considering such additional constraints.
> It would be very interesting to discuss the theoretical motivation why researchers before thought
> that such end-to-end training is a suitable approach. If this is based on theoretical assumptions
> about, for example, transformations of feature spaces which fall out generically from end-to-end
> training of deep architectures that make additional intermediate layer constraints redundant?
> If this should be the case, the presented work would provide evidence against the validity of these
> theoretical assumptions, which would be potentially an interesting general theoretical insight.

---

> > ### Author Response · Authors · 2025-08-06
> >
> > We thank the reviewer for raising this important theoretical point. As noted, it is indeed expected that sufficiently flexible architectures trained end-to-end can fit neural responses. However, we highlight two key reasons why hierarchical readouts were not necessary in prior work but are essential in our case. First, previous studies have focused almost exclusively on modeling a single area, V1 (e.g. [1-4]). Second, evaluations have largely been restricted to single-neuron prediction accuracy. In contrast, our study targets a broader and more biologically meaningful objective: capturing not only single-neuron responses but also structured variation across multiple areas of the visual hierarchy. To this end, we move beyond neuron-level metrics and evaluate models based on population-level properties such as object discriminability and inter-area response structure. Critically, we find that while single-neuron accuracy remains relatively unchanged, our hierarchical readout substantially improves population-level metrics and successfully recovers empirical patterns that standard end-to-end models fail to reproduce.
> >
> > Regarding the reviewer’s theoretical question: we expect that a sufficiently deep and well-trained end-to-end model could learn internal representations that support the same hierarchical structure as our model. However, doing so would require significantly more data and model capacity. Our hierarchical readout serves as an inductive bias that makes this structure easier to learn with less data, and in models of more manageable size. Rather than contradicting prior theoretical assumptions, our findings complement them by demonstrating that appropriate architectural constraints can improve sample efficiency and alignment with biological organization. In this sense, our results provide empirical evidence that intermediate constraints are not redundant in practice, even if they are theoretically unnecessary in the limit of infinite data and capacity. We believe this insight opens promising theoretical and practical directions for modeling neural systems, highlighting the importance of inductive biases in achieving biologically faithful representations.
> >
> > [1] Klindt et al. “Neural system identification for large populations separating “what” and “where”.” Advances in neural information processing (2017).
> > [2] Sinz et al. "Stimulus domain transfer in recurrent models for large scale cortical population prediction on video." Advances in neural information processing systems  (2018).
> > [3] Turishcheva et al. “Retrospective for the dynamic sensorium competition for predicting large-scale mouse primary visual cortex activity from videos” arXiv (2024)
> > [4] Lurz et al. “Generalization in data-driven models of primary visual cortex.” ICLR (2021)

---

> > > ### Comment · Reviewer_vF7y · 2025-08-06
> > > **Thanks for comments.**
> > >
> > > I thank the authors for their insightful comments. I fully agree that the additional constraints make learning
> > > on limited data faster and more efficient. I am not sure if learning even bigger models end-to-end will really necessarily lead to
> > > models that reflect internally the structure / invariances at different levels of the brain. But probably nobody knows this :-)
> > > Many thanks for your answer which is satisfying for me.

---

### Official Review · Reviewer_vQD3 · 2025-07-02

**Clarity:** 2
**Significance:** 3
**Originality:** 3
**Rating:** 5
**Confidence:** 4

**Summary:**

The authors study digital twins, networks trained to predict neural activity in the brain, to understand whether these networks learn invariant object recognition using the same low-level and high-level principles as those used in the real visual cortex. They reference previous studies that identify properties of object recognition in visual cortex and find that only low-level properties are present in digital twins.

**Questions:**

•	Use of Readout: Can the readout for RL and LM be treated the same way? I may be misunderstanding, but it seems RL and LM have the same readout depth.
•	Different Neuron types: The digital twin treats every neuron as an identical readout, ignoring the huge diversity of excitatory vs. inhibitory cell types and the laminar structure within each cortical area. Does this have an effect on the results?

**Ethical Concerns:**

["NO or VERY MINOR ethics concerns only"]

**Final Justification:**

The thorough response from the authors and their answers to the questions I raised have convinced me that this manuscript is suitable for publication.

**Limitations:**

•	Addressed adequately in the paper.

**Paper Formatting Concerns:**

No formatting concerns

**Quality:**

2

**Strengths And Weaknesses:**

Strengths:
•	Novelty: The paper identifies gaps in digital twins and analyzes where digital models fail to accurately reproduce well-known brain phenomena.
•	Empirical gaps: The paper demonstrates that digital twins are unable to reproduce certain phenomena, such as area-to-area relationships or hierarchical representations.
•	Evaluation: The paper uses benchmarks and evaluations that are extensive and justified.
Weaknesses:
•	Part of the reason for using a linear readout rather than a hierarchically designed readout is to enforce a simplicity bias, i.e., to ensure that whatever is decoded is not due to the complexity of the decoder. How do the authors propose that their hierarchical readout design avoids complexity bias in this way?
•	The design of the convolutional architecture seems odd to me for drawing conclusions. Even Wang et. al. (2025), a paper they cite, has a much larger and more extensive network architecture. Would every “digital twin” need a hierarchical readout? How wide should the applicability be?
o	Furthermore, I am slightly confused about the network depth experiments. It would be helpful to know how many parameters were actually added to the network when varying network depth.
•	There are some methodological/design choices that are perhaps justified in other digital twins papers, but in my opinion, change the narrative presented in this paper, and I feel these should be addressed.
o	Bypassing Subcortical Regions: The authors rightfully acknowledge that they do not consider the retina or LGN. Although other studies also omit this region, I believe that ignoring where invariances begin to emerge weakens the claim that their hierarchical readout model is very useful. Perhaps representations in this region contain enough hierarchical information to make their own readout less competitive. I think this could benefit from further discussion.
o	I am not sure RL and LM can have the same readout; see questions.
o	Use of Calcium Data: Could the paper discuss how results would change if a neural recording measure other than calcium data was used?

---

> ### Author Rebuttal · Authors · 2025-07-30
>
> We thank the reviewer for the thoughtful and thorough review. Below, we respond point by point to each concern and will incorporate clarifications and additional analyses in the revised paper.
>
> ---
>
> **1. Complexity bias in hierarchical readout**
> Our hierarchical readout is composed exclusively of simple linear projections followed by a rectification step to ensure non-negative outputs, consistent with biological firing rate constraints. It does not include any additional nonlinear components such as multilayer perceptrons (MLPs), nor does it increase the parameter count relative to standard readouts used in prior work (e.g., [1,2]). This ensures that the function class of the decoder remains unchanged and maintains the simplicity bias that has historically motivated digital twin architectures. The novelty of our approach lies in the anatomical organization of the readout: rather than pooling all brain areas into a shared readout layer, as is standard, we assign each area its own dedicated readout layer based on its hierarchical position in the cortical structure. Importantly, the term "hierarchical" in our method refers solely to this anatomical alignment and does not imply added computational complexity.
>
> [1] Sinz, Fabian, et al. NeurIPS (2018)
> [2] Willeke, Konstantin F., et al. arXiv (2022)
>
> ---
>
> **2. Architecture design and applicability**
> Our choice of a relatively compact convolutional architecture was guided by a balance between computational tractability and the need for sufficient representational power. Importantly, this architecture has been used successfully in prior digital twin work and provides strong baseline performance [1]. To go beyond earlier baselines, we extended the architecture from 4 to 8 layers and incorporated a hierarchical readout. Combined with the full MICrONS dataset (all available sessions), this version achieves performance comparable to the model of Wang et al. (2025), which, to our knowledge, is the current state of the art in modeling mouse visual cortex responses according to standard single neuron metrics. Importantly, our model is significantly smaller (number of parameters: 20,824,610 ours, vs 58,915,491 Wang 2025), is trained on a smaller number of neurons (115k ours, vs 135k Wang 2025) and training set (minutes of video: 692 ours, vs 900 Wang 2025). Despite this, our hierarchical model achieves similar performance on key single-neuron metrics (median CCnorm in session: [0.60–0.72] ours vs [0.58–0.76] Wang 2025).
> To directly assess whether hierarchical readouts are essential for success in this context, we adapted the Wang et al. model to our population-level evaluation framework and re-ran the core experiments from Hoeller et al. and Froudarakis et al. In both experiments, the Wang model performs significantly worse than our hierarchical model (see Table 1 and 2 below). Specifically, while it captures the broad difference between V1 and higher areas (i.e., lower discriminability in V1), it fails to reproduce distinctions across higher order areas (i.e., AL vs. RL/LM), predicting nearly identical performance. This is inconsistent with empirical findings that show a clear AL advantage. In contrast, our hierarchical model replicates both the V1-vs-higher-area difference and correctly ranks AL above RL and LM, aligning closely with experimental data.
> These findings support our central claim: hierarchical readouts provide a useful inductive bias that helps even small models capture key aspects of the visual hierarchy. Increasing model size or training data alone does not recover these features. While larger models may boost overall performance, our results suggest they still benefit from anatomically structured readouts.
>
> **Table 1. Difference in classification accuracy between areas (Hoeller)**
> | Model | AL–V1 | AL–RL | AL–LM |
> |-------|--------|--------|--------|
> | Our Hierarchical Model | 0.072 ± 0.0046 (p = 4.77e−15) | 0.027 ± 0.0050 (p = 2.27e−06) | 0.028 ± 0.0049 (p = 1.01e−06) |
> | Wang et al. (2025) | 0.043 ± 0.0079 (p = 4.09e−05) | –0.013 ± 0.0096 (0.720, n.s.) | 0.003 ± 0.0078 (0.209, n.s.) |
>
> **Table 2. Difference in object discriminability between areas (Froudarakis)**
> | Model | AL–V1 | AL–RL | AL–LM |
> |-------|--------|--------|--------|
> | Our Hierarchical Model | 0.0585 ± 0.0038 (p = 8.33e−14) | 0.0195 ± 0.0040 (p = 7.93e−06) | 0.0387 ± 0.0036 (p = 6.63e−12) |
> | Wang et al. (2025) | 0.0234 ± 0.0066 (p = 0.0061) | –0.0084 ± 0.0059 (0.312, n.s.) | 0.0023 ± 0.0068 (0.237, n.s.) |
>
> [1] Turishcheva et al., arXiv (2024)
>
> ---
>
> **3. Network depth and parameter count**
> Varying the depth of the convolutional core affects the number of internal parameters, but the number of readout parameters is kept constant across all model variants (including those with hierarchical and non-hierarchical readouts). This allows us to isolate the effect of core architecture depth without conflating it with decoder complexity.
>
> | Model Variant | Core Parameters | Readout Parameters |
> |---------------|------------------|---------------------|
> | Baseline (4 layers) | 382,512 | 17,396,557 |
> | Baseline (6 layers) | 2,432,816 | 17,396,557 |
> | Baseline (8 layers) | 10,629,424 | 17,396,557 |
> | Hierarchical (8 layers) | 3,417,392 | 17,396,557 |
>
> ---
>
> **4. Methodological and design choices**
> In Supp. Section A we provide a rationale for our architectural and design choices, referencing prior work that supports each of them. Our goal was to follow established practices while introducing the hierarchical readout as the key novel component. We would greatly appreciate it if the reviewer could point out any specific design choices they feel were insufficiently justified or that significantly alter the narrative, so that we can address them more directly and clarify our intentions in the revised manuscript.
>
> ---
>
> **5. Bypassing subcortical regions**
> Our current focus is on explaining the increase in invariance from V1 to higher areas like AL, as shown in prior work (e.g., Froudarakis et al., Hoeller et al.). While the retina and LGN may contribute to early visual processing, digital twins without a hierarchical readout fail to capture the functional gradient across cortical areas, even if some invariance arises earlier.
>
> ---
>
> **6. RL and LM readout depth**
> Our original hierarchical readout design was informed by both performance optimization and anatomical constraints. V1 performance was best captured using a readout from the 6th layer of our convolutional core. To model downstream areas, we extended the core and placed readouts at progressively deeper layers based on anatomical hierarchy scores from [1], as shown in Fig. 1. This hierarchy follows the order V1 < LM < RL < AL. Since the separation between LM and RL is small, we approximated their depth as equivalent and used a shared readout at layer 7. We acknowledge this as a simplification.
> To assess the validity of this approximation and directly address the reviewer’s concern, we trained an additional model variant that explicitly separates LM and RL readout paths. Inspired by the dorsal/ventral stream distinction in mouse visual cortex [2], this architecture splits into two branches after layer 6. The dorsal stream targets RL and AL with readouts at layers 7 and 8, respectively. The ventral stream targets LM with a separate readout at layer 7. Due to time constraints, we evaluated this variant only in the original six sessions, not the full set of 14 used in the new analysis referenced elsewhere in the response. The model yields improved performance across all areas (Table 1), and preserves the global structure of inter-area relationships (Table 2, 3). This confirms that the hierarchical readout framework is not sensitive to the specific treatment of LM and RL depth, and that our findings hold under a more anatomically refined design.
>
> **Table 1. CCnorm per area for 8-layer models**
> | Model | V1 | LM | RL | AL |
> |--------|------|------|------|------|
> | Hierarchical Readout (8-layer) | 0.601 ± 0.002 | 0.553 ± 0.001 | 0.554 ± 0.002 | 0.602 ± 0.003 |
> | Two-streams Hierarchical Readout | 0.675 | 0.622 | 0.625 | 0.636 |
>
> **Table 2. Classification accuracy and Kendall's Tau (Hoeller)**
> | Model | V1 | LM | RL | AL | Tau |
> |--------|------|------|------|------|------|
> | Hierarchical Readout (8-layer) | 0.183 ± 0.004 | 0.213 ± 0.006 | 0.203 ± 0.006 | 0.231 ± 0.003 | 0.56 ± 0.07 |
> | Two-streams Hierarchical Readout | 0.180 ± 0.005 | 0.210 ± 0.007 | 0.207 ± 0.006 | 0.238 ± 0.004 | 0.82 ± 0.11 |
>
> **Table 3. Mutual information and Kendall's Tau (Froudarakis)**
> | Model | V1 | LM | RL | AL | Tau |
> |--------|------|------|------|------|------|
> | Hierarchical Readout (8-layer) | 0.365 ± 0.005 | 0.378 ± 0.004 | 0.387 ± 0.005 | 0.402 ± 0.004 | 0.31 ± 0.07 |
> | Two-streams Hierarchical Readout | 0.376 ± 0.006 | 0.387 ± 0.004 | 0.430 ± 0.003 | 0.425 ± 0.006 | 0.11 ± 0.11 |
>
> [1] Harris et al., Nature (2019)
> [2] D’Souza et al., Nat. Commun. (2022)
>
> ---
>
> **7. Neuron types**
> Our decision to treat all neurons identically reflects the limitations of the MICrONS dataset, which contains only excitatory neuron recordings. Including inhibitory neurons could enhance model training by providing additional information about local circuit dynamics. While prior work shows that model performance improves with more neurons [1], inhibitory neurons are typically less selective, so their per-neuron contribution may be smaller. Importantly, we show that single-neuron performance does not reliably predict a model’s ability to capture the functional hierarchy. Moreover, the experimental data we aimed to reproduce also focused solely on excitatory neurons. Thus, this simplification does not compromise our main findings. Extending the approach to include both excitatory and inhibitory populations is a valuable future direction but beyond the scope of this study.
>
> [1] Lurz et al., ICLR (2021)

---

> ### Comment · Reviewer_vQD3 · 2025-08-05
>
> The authors have answered all my questions and addressed my concerns, so I am increasing the rating to 5.

---

> > ### Comment · Area_Chair_1Wu6 · 2025-08-06
> >
> > Dear reviewer,
> >
> > Thank you for your contribution to the review process. I note that you mentioned an increase in your score following the authors' rebuttal. Could you please also update the score in your initial review to reflect this change?
> >
> > Best,
> > Area Chair

---

### Official Review · Reviewer_MMjt · 2025-07-02

**Clarity:** 2
**Significance:** 3
**Originality:** 3
**Rating:** 4
**Confidence:** 4

**Summary:**

This manuscript demonstrates that while previous artificial neural network models of the mouse visual cortex can predict single-neuron activity, they fail to replicate the hierarchical emergence of invariant object representations across different visual areas. To address this limitation, the authors introduce a biologically-inspired hierarchical readout scheme that maps different visual areas to distinct depths of a shared neural network, mirroring their anatomical organization. This anatomically constrained model is shown to significantly outperform standard architectures by successfully capturing key experimental findings, such as the progressive increase in both receptive field size and invariant object decoding accuracy along the cortical hierarchy. The authors conclude that incorporating anatomical information provides a powerful inductive bias that enables computational models to better capture the population-level transformations underlying complex brain functions.

**Questions:**

**Major concerns**

1. The proposed hierarchical readout model maps areas V1, LM/RL, and AL to the 6th, 7th, and 8th layers of an 8-layer network, respectively. What is the specific rationale for assigning V1 to a relatively deep layer like the 6th? Given that early visual areas are typically associated with the early layers of deep learning models, how can you disentangle the impact of this design choice—for instance, the performance gains from the increased expressivity of a deeper network—from the effect of the hierarchical readout scheme itself?

2. The core contribution of this study is its ability to capture population-level phenomena that were missed by previous models. However, it remains unclear whether the hierarchical model, in the course of better matching population-level orderings, can still reproduce the fine-grained, single-neuron properties that were the focus of prior research. The claims would be significantly strengthened by demonstrating that the proposed model not only improves the prediction of the population hierarchy but also maintains or even exceeds the precision of existing models in predicting single-neuron response characteristics.

3. The proposed architecture features different brain areas reading out in parallel from different depths of a single, shared core network. This abstracts away from the brain's cascaded processing, where the output of V1, for instance, is the input for intrinsic computations within LM before projecting further. The current model lacks these area-specific, cascaded stages (e.g., Core_V1 → Module_LM → Module_AL). How much biological realism is lost in this simplification?

4. The model is trained on data from the MICrONS dataset, which includes only excitatory neurons. However, inhibitory interneurons are critical for shaping receptive fields, controlling gain, and creating temporal dynamics within the cortex. By excluding this entire class of neurons, the model is inherently incomplete. The observed functional hierarchy might not be a product of feedforward excitatory connections alone, but could be critically sculpted by local inhibitory circuits, especially given that mechanisms like selectivity and invariance are often attributed to inhibition.

Liu et al., "Intervening inhibition underlies simple-cell receptive field structure in visual cortex," Nat. Neurosci., 2009

Larisch et al., "Sensory coding and contrast invariance emerge from the control of plastic inhibition over emergent selectivity," PLoS Comp. Biology, 2021

5. As you note in your discussion, Hoeller et al. (2024) proposed that the brain maintains equivariant representations in addition to invariant ones, whereas a DNN such as AlexNet achieves invariance by suppressing equivariant responses. However, other studies suggest different aspects of object invariance in DNNs. For example, Farzmahdi et al. (2024) suggest that brain-like, mirror-symmetric viewpoint tuning emerges in AlexNet. In addition, Cheon et al. (2022) demonstrate that hierarchically increasing object invariance—alongside specific (equivariant) coding—emerges even in an untrained AlexNet. These studies highlight that brain-like invariance tuning, specifically across a hierarchy, may also emerge in artificial neural network models. Comparing the digital twin's hierarchical representation with these previous models may bridge the gap between data-driven, task-optimized, and structure-inherent approaches to modeling the visual cortex.

Hoeller et al., “Bridging tuning and invariance with equivariant neuronal representations,” bioRxiv, 2024

Farzmahdi et al., “Emergence of brain-like mirror-symmetric viewpoint tuning in convolutional neural networks,” eLife, 2024

Cheon et al., “Invariance of object detection in untrained deep neural networks,” Front. Comput. Neurosci., 2022

**Minor suggestions**

1. A point of confusion arises from the description of the experimental ground truth on page 8. The text states that "...experimentally RL ranks very slightly above LM in RF size, cf. Fig. 1B (line 276)" to explain the Kendall's τ score. However, this appears to be inconsistent with the data presented in Figure 1B itself. Both the figure's caption, which specifies a hierarchy of V1<RL<LM<AL, and a visual inspection of the plot suggest that LM has a larger receptive field area than RL. This discrepancy could create confusion for the reader and impacts the interpretation of the model's alignment with the experimental findings. Clarification on this point would be helpful.

**Ethical Concerns:**

["NO or VERY MINOR ethics concerns only"]

**Final Justification:**

This work provides valuable insights into modeling hierarchical information processing in the visual cortex and understanding how populational information is transformed across its layers. Therefore, I support the acceptance of this paper, as in my original review.

**Limitations:**

The purely feedforward architecture omits the critical roles of feedback and recurrent connections, which are essential for complex visual processing and representation in the biological brain.

**Paper Formatting Concerns:**

No paper formatting concerns

**Quality:**

3

**Strengths And Weaknesses:**

**Strengths**

The anatomically-inspired hierarchical readout successfully enables the model to capture the emergent hierarchy of invariant object representations across the visual cortex, a feat that standard architectures fail to achieve.

**Weaknesses**

The model's oversimplified architecture treats functionally distinct areas, such as LM and RL, identically, causing it to fail in reproducing key experimental findings related to the unique role of area RL.

---

> ### Author Rebuttal · Authors · 2025-07-30
>
> We thank the reviewer for the thoughtful and thorough review. Below, we respond point-by-point to each of the reviewer’s comments and clarify the steps taken to address the concerns raised.
>
> ---
>
> ### **Major Concerns**
>
> ---
>
> **1. Rationale for mapping V1 to layer 6, and disentangling hierarchical readout vs. increased expressivity due to depth**
>
> Our assignment of V1 to layer 6 was based on empirical validation: performance improved when moving from layer 4 to 6, with no significant gain beyond that (Fig 3). We therefore selected the simplest effective configuration that yielded high performance. It’s also worth noting that while V1 is anatomically early in the cortical hierarchy, its activity results from multiple preceding stages of nonlinear processing. In fact, several recent studies have shown that retinal circuits alone can be well-modeled by multi-layer deep networks [1]. Additionally, V1 responses to thalamic inputs are shaped by complex recurrent dynamics. Given this, it is not surprising that deeper layers in artificial neural networks are required to best model V1 responses.
>
> We agree that disentangling the effect of hierarchical readout from increased depth is crucial. This is addressed in Fig 3, which compares models with and without hierarchical readouts at matched depth, and in Fig 4, which controls for depth while varying readout organization. These analyses demonstrate that the performance gains stem from the anatomically inspired hierarchy rather than from depth increase.
>
> [1] Tanaka H, et al. *From deep learning to mechanistic understanding in neuroscience: the structure of retinal prediction.* Advances in Neural Information Processing Systems (2019)
>
> ---
>
> **2. Does the model preserve single-neuron precision while capturing population-level structure?**
>
> We agree that strong single-neuron predictive accuracy remains important. To address this, we expanded training to include all 14 MICrONS sessions (rather than only 6 as in our initial submission). With the full dataset  (Table 1), our 8-layer hierarchical readout model matches the performance of depth-matched models with non-hierarchical readouts, in which all areas are predicted from the final layer. Note that, while both models use the same number of parameters in the readout (17,396,557), the non-hierarchical model has a substantially larger core (10,629,424 parameters) compared to the hierarchical variant (3,417,392 parameters), resulting in more total parameters. Despite this added capacity, overall performance remains equivalent across models. This performance is comparable to the current state-of-the-art (Wang et al., 2025). When analyzing performance by area (Table 2), we find that the hierarchical model improves accuracy in AL, the top of the functional hierarchy, compared to a depth-matched baseline. Performance for V1, LM, and RL remain unchanged. These findings confirm that hierarchical readout preserves, and in some cases improves, single-neuron prediction accuracy while enabling better alignment with population-level structure.
>
> **Table 1.** Average single-neuron predictive performance (correlation) per model.
> Values are reported as the median normalized correlation coefficient (CC_norm) across all neurons (mean ± SEM across three random seeds), along with the range of per-session medians. Wang et al. (2025) is excluded from the average due to the lack of per-neuron data.
> | Model                           | Median CC_norm     | Median CC_norm per session |
> |--------------------------------|---------------------|-----------------------------|
> | Hierarchical (8 layers, hierarchical readout) | 0.648 ± 0.010       | [0.60 – 0.72]               |
> | Baseline (8 layers, last-layer readout)  | 0.648 ± 0.020       | [0.61 – 0.73]               |
> | Wang et al. (2025)             | not available       | [0.58 – 0.76]               |
>
> **Table 2.** Median CC_norm per area for 8-layer models with and without hierarchical readout.
> Values are reported as mean ± SEM across three random seeds. Data from Wang et al. (2025) are not included due to the unavailability of area-level measurements.
>
> | Model                           | V1            | LM            | RL            | AL            |
> |--------------------------------|---------------|---------------|---------------|---------------|
> | Hierarchical Readout (8-layer) | 0.662 ± 0.002 | 0.619 ± 0.001 | 0.604 ± 0.002 | 0.615 ± 0.003 |
> | Baseline (last-layer readout)  | 0.663 ± 0.002 | 0.616 ± 0.002 | 0.602 ± 0.001 | 0.600 ± 0.002 |
>
> ---
>
> **3. Biological realism and the lack of cascaded processing**
>
> In our current architecture, responses in each visual area are computed from the output of a shared convolutional core, with parallel linear readouts from different layers. The core itself is hierarchical and cascaded in the sense that deeper layers (e.g., layer 8 for AL) build on earlier ones (e.g., layer 7 for LM/RL). However, we do not explicitly model inter-area communication as it occurs in biological circuits, where the output of one area directly informs the next. This is a simplification made to accommodate current data-driven modeling approaches.
>
> Explicitly routing activity from one area’s readout to another could in principle be more biologically realistic. However, we do not expect this to substantially alter the results. The transformations between layers in our shared core already approximate such computations and can be learned directly from the data. Moreover, because we observe only a subset of neurons in each area, constraining downstream areas to use only observed upstream activity might limit performance. In contrast, accessing shared internal representations allows the model to capture a more complete and abstract embedding of the population response, improving generalization.
>
> We will clarify this modeling choice in the revised manuscript and acknowledge the potential value of future work exploring modular architectures with explicit inter-area pathways.
>
> ---
>
> **4. Exclusion of inhibitory neurons**
>
> The MICrONS dataset used in this study contains functional recordings exclusively from excitatory neurons, and our model is trained and evaluated based solely on that activity. This is consistent with the experimental data we aim to replicate, specifically the studies by Hoeller et al. and Froudarakis et al., which also report responses from excitatory neurons only.
>
> Therefore, the functional properties and hierarchical organization we model arise from excitatory populations, which aligns with the scope of both the available data and the experiments we seek to reproduce. While inhibitory interneurons are known to influence receptive fields, gain control, and temporal dynamics, our model is not intended to mechanistically capture those interactions. Instead, it is a functional model that aims to learn statistical regularities in the observed excitatory responses.
>
> Including inhibitory neurons, on the other hand, could potentially improve training by providing additional information about local circuit dynamics. However, because inhibitory neurons are typically less selective, their per-neuron contribution to predictive performance may be limited.
>
> Finally, we emphasize that our goal is not to maximize single-neuron prediction but to reproduce the functional hierarchy observed in population-level data. Our results show that training directly on excitatory responses is sufficient to recover this structure.
>
> ---
>
> **5. Brain-like invariance and equivariance in DNNs**
>
> We agree with the reviewer on the importance of emergent invariance and equivariance in artificial neural networks and appreciate the opportunity to clarify how our study relates to prior findings. Farzmahdi et al. (2024) and Cheon et al. (2022) offer complementary insights, demonstrating that task-driven models, such as AlexNet, can exhibit mirror-symmetric viewpoint tuning, while the same randomly initialized architectures already possess invariance properties. These findings suggest that the emergence of invariance and equivariance is shaped not only by network architecture, but also by the training objective and the dataset.
>
> Although our study does not directly probe invariance or equivariance, it builds upon these insights by comparing model outputs to experimental data from Hoeller et al. and Froudarakis et al. Specifically, we assess population-level representational changes across hierarchical cortical areas using digital twin models. Our objective is to examine how invariance to object transformations, such as viewpoint, scale, and position, is distributed across cortical regions, and how this invariance emerges with anatomical hierarchy.
>
> To address the limitations of standard models, which typically use a shared readout from the final layer and can match single-unit responses but fail to capture hierarchical population geometry, we introduce an anatomically-inspired hierarchical readout. This strategy enables the model to capture higher-order response patterns across brain areas by allowing structured variation in how populations access shared features.
>
> We recognize that a systematic dissection of invariance and equivariance across architectures, training regimes, and biological constraints represents an important future direction. We will revise the discussion section to reflect these connections more explicitly and emphasize how our findings relate to and complement the cited studies.
>
> ---
>
> ### **Minor Points**
>
> - **RL vs. LM ranking in receptive field size (Figure 1B)**
>   Thank you for catching this inconsistency. This was a typo in the original manuscript text. The correct experimental ranking is V1 < RL < LM < AL, as shown in the figure. We will correct this in the revised version.

---

> ### Comment · Reviewer_MMjt · 2025-08-04
>
> Thank you to the authors for their detailed responses and clarifications. The additional results on single‐neuron–level predictions address my concerns. I also believe that the discussion of emergent invariance and equivalence in hierarchical models, and its connection to prior work, will be valuable for the field. Although I have minor concerns regarding the model design and its biological plausibility, I believe this work offers valuable insights for modeling biological vision and object representations. Therefore, I support the acceptance of this paper.

---

### Official Review · Reviewer_JMCZ · 2025-07-02

**Clarity:** 4
**Significance:** 3
**Originality:** 2
**Rating:** 5
**Confidence:** 4

**Summary:**

This work examines whether data-driven models of neural responses in mouse visual cortex reproduce observations from the brain regarding invariance of object representations. A baseline model predicting all units from the same layer achieves good single-unit correlations, but fails to capture the hierarchy of invariant object representations across brain areas. This shortcoming persists under several changes of the backbone architecture. Finally, reading out each brain area from its own model layer, according to its position in the hierarchy along the visual pathway, leads to a model that is more consistent with brain data at the population level while retaining single-unit performance.

**Questions:**

1. Perhaps the anatomical hierarchy score shown in Fig. 2 could be slightly better explained. I found the x-axes there a little confusing.
2. Small typos in lines 82, 247

**Ethical Concerns:**

["NO or VERY MINOR ethics concerns only"]

**Final Justification:**

The thorough discussion of the questions I've raised has reinforced my position that this manuscript is suitable for publication.

**Limitations:**

yes

**Paper Formatting Concerns:**

I see no formatting concerns beyond the typos pointed out above.

**Quality:**

3

**Strengths And Weaknesses:**

**Strengths**

The logical flow of the experiments and conclusions is strong and well-presented. The experiments themselves properly support the claims made in the manuscript. Some further ways of characterizing global representational properties are given in the limitations section, but the ones that are tested in the experiments are convincing in their own right. Further, it is interesting to see that the models differ strongly in their ability to capture global geometries, while single-unit predictivity is very similar. It is an important message to the field to look beyond single-unit correlations.

**Weaknesses**

The only issue I see lies in the novelty of the approach, as the finding that deeper ANN layers correspond more to higher-level visual areas is well established in the literature on task-driven models of the brain (e.g., [1], [2], [3]). Therefore, a data-driven model with a corresponding structure can be considered a somewhat minor advancement. The discussions in the manuscript focus heavily on data-driven models and gloss over this fact – I would recommend that a small section should be dedicated to exploring how the findings presented here relate to task-driven models.

Nevertheless, the work is well done and I recommend it be accepted for publication.

[1] Performance-optimized hierarchical models predict neural responses in higher visual cortex

[2] Deep Neural Networks Rival the Representation of Primate IT Cortex for Core Visual Object Recognition

[3] Deep Neural Networks Reveal a Gradient in the Complexity of Neural Representations across the Brain’s Ventral Visual Pathway

---

> ### Author Rebuttal · Authors · 2025-07-30
>
> We thank the reviewer for their thoughtful and encouraging comments. We appreciate the recognition of the importance of moving beyond single-unit correlations to study population-level representations.
>
> **Novelty and Relation to Prior Work**
>
> Regarding the concern about novelty, we would like to clarify a key distinction between findings in primates and mice that underpins the contribution of our work. In primates, Yamins et al. [1] demonstrated that performance-optimized deep networks trained on object recognition exhibit a clear hierarchical correspondence with the ventral visual stream: early layers best predict V1, middle layers align with V4, and deeper layers correspond to IT cortex. This provided strong evidence that task-driven models can approximate the functional hierarchy of primate vision.
>
> However, when the same approach was applied to the mouse visual system, Cadena et al. [2] found no evidence of such hierarchical alignment. While CNN features outperformed classical subunit energy models, randomly initialized networks performed comparably to ImageNet-trained models in predicting mouse neural responses. Critically, all visual areas studied (including V1, LM, AL, and RL, which we also examine in our work) mapped similarly across layers, showing no consistent progression from lower to higher areas.
>
> More recently, Nayebi et al. [3] extended this analysis by systematically varying model architectures, training objectives, and input statistics. They found that certain unsupervised objectives, particularly contrastive learning, outperformed both random and supervised models in predicting mouse neural activity. While these models consistently mapped V1 to early layers and higher visual areas such as LM, AL, and RL to intermediate layers, they failed to differentiate among these higher-order areas. As a result, they cannot account for the empirical findings (e.g., Froudarakis et al., Hoeller et al., both examined in this manuscript) that reveal clear differences in object recognition performance across these regions (see Fig. 1 of our manuscript).
>
> In this context, we believe our findings are both surprising and novel. Instead of relying on predefined tasks or training objectives, we use a *data-driven approach*, training a model directly on neural responses. The resulting *digital twin* successfully recovers meaningful distinctions between visual areas, particularly in their object recognition performance. By allowing each area to be read out from an individually matched model layer, we reveal a functional hierarchy that aligns with observed behavioral and representational differences, something that task-driven models have not achieved. This suggests that models trained directly on neural data may be better suited to capturing the computational organization of the mouse visual cortex than those relying on external tasks or objectives.
>
> Following the reviewer’s suggestion, we will add a brief section to clarify the distinction between task-driven and data-driven approaches and to acknowledge the broader relevance of such comparisons. We hope this helps reinforce the novelty and significance of our findings.
>
> **Clarification of Anatomical Hierarchy Score**
>
> In Figure 2B, the x-axis shows the anatomical hierarchy score introduced by Harris et al. [4], which assigns a continuous value to each brain area reflecting its relative position in a cortical–thalamic hierarchy. This score is computed by analyzing the laminar termination patterns of axonal projections between areas. Each projection is classified as either feedforward or feedback based on its laminar pattern, and an optimization procedure finds the mapping that yields the most self-consistent global hierarchy.
>
> The initial score of each area is based on the average directionality of its incoming and outgoing projections, and is refined via iterative updates that account for the positions of source and target areas. This method captures the idea that, for example, feedback projections from higher to lower areas tend to terminate in deep layers, while feedforward projections from lower to higher areas terminate more superficially.
>
> In our manuscript, we introduce this hierarchy score briefly in Figure 1A and use it throughout the text, including as the independent variable in Figure 2B, where we relate it to receptive field size, object decoding accuracy, and discriminability. We will expand the description of this score in the revised manuscript.
>
>
> **Minor Comments**
>
> Typos on lines 82 and 247 will be corrected in the revised manuscript.
>
> **References**
>
> [1] Yamins, D. and DiCarlo, J. *Using goal-driven deep learning models to understand sensory cortex.* *Nature Neuroscience* (2016)
> [2] Cadena, S.A., et al. *How well do deep neural networks trained on object recognition characterize the mouse visual system?* *NeurIPS* (2019)
> [3] Nayebi, A., et al. *Mouse visual cortex as a limited resource system that self-learns an ecologically-general representation.* *PLOS Computational Biology* (2023).
> [4]: Harris et al. “Hierarchical organization of cortical and thalamic connectivity”. Nature (2019)

---

> > ### Comment · Reviewer_JMCZ · 2025-08-01
> >
> > Thank you for your response and interesting discussion. As indicated in my original review, I do believe in this work's merit and recommend it be accepted for publication.

---

### Decision · Program_Chairs · 2025-09-17

**Decision:**

Accept (poster)

**Comment:**

This work examines digital twin models of the mouse visual system: ANNs trained to predict visual neural responses. The authors specifically investigate whether these digital twin models can reproduce population-level properties of visual responses, extending beyond the models' capability to explain individual neuron responses.

The reviewers unanimously agreed that this paper represents a very interesting and valuable contribution that advances our understanding of how to build end-to-end trained ANNs for modeling visual systems. I share this assessment and am convinced of both the scientific value of this work and its methodological rigor. Therefore, I recommend acceptance.